# CONTEMPLATING REAL-WORLD OBJECT CLASSIFICATION

**Ali Borji**
HCL America
aliborji@gmail.com

## ABSTRACT

Deep object recognition models have been very successful over benchmark datasets such as ImageNet. How accurate and robust are they to distribution shifts arising from natural and synthetic variations in datasets? Prior research on this problem has primarily focused on ImageNet variations (e.g., ImageNetV2, ImageNet-A). To avoid potential inherited biases in these studies, we take a different approach. Specifically, we reanalyze the ObjectNet dataset[1] recently proposed by Barbu et al. containing objects in daily life situations. They showed a dramatic performance drop of the state of the art object recognition models on this dataset. Due to the importance and implications of their results regarding the generalization ability of deep models, we take a second look at their analysis. We find that applying deep models to the isolated objects, rather than the entire scene as is done in the original paper, results in around 20-30% performance improvement. Relative to the numbers reported in Barbu et al., around 10-15% of the performance loss is recovered, without any test time data augmentation. Despite this gain, however, we conclude that deep models still suffer drastically on the ObjectNet dataset. We also investigate the robustness of models against synthetic image perturbations such as geometric transformations (e.g., scale, rotation, translation), natural image distortions (e.g., impulse noise, blur) as well as adversarial attacks (e.g., FGSM and PGD-5). Our results indicate that limiting the object area as much as possible (i.e., from the entire image to the bounding box to the segmentation mask) leads to consistent improvement in accuracy and robustness. Finally, through a qualitative analysis of ObjectNet data, we find that i) a large number of images in this dataset are hard to recognize even for humans, and ii) easy (hard) samples for models match with easy (hard) samples for humans. Overall, our analyses show that ObjecNet is still a challenging test platform for evaluating the generalization ability of models. Code and data are available at https://github.com/aliborji/ObjectNetReanalysis.git[2].

## 1 INTRODUCTION

Object recognition[3] can be said to be the most basic problem in vision sciences. It is required in the early stages of visual processing before a system, be it a human or a machine, can accomplish other tasks such as searching, navigating, or grasping. Application of a convolutional neural network architecture (CNN) known as LeNet (LeCun et al., 1998), albeit with new bells and whistles (Krizhevsky et al., 2012), revolutionized not only computer vision but also several other areas. With the initial excitement gradually damping, researchers have started to study the shortcomings of deep models and question their generalization ability. From prior research, we already know that CNNs: **a**) lack generalization to out of distribution samples (e.g., Recht et al. (2019); Barbu et al. (2019); Shankar et al. (2020); Taori et al. (2020); Koh et al. (2020)). Even after being exposed to many different instances of the same object category, they fail to fully capture the concept. In stark contrast, humans can generalize from only few examples (a.k.a few-shot learning), **b**) perform poorly when applied to transformed versions of the same object. In other words, they

---

[1] https://objectnet.dev/

[2] See https://openreview.net/forum?id=Q4EUywJIkqr for reviews and discussions. A preliminary version of this work has been published in Arxiv (Borji, 2020).

[3] Classification of an object appearing lonely in an image. For images containing multiple objects, object localization or detection is required first.

are not invariant to spatial transformations (e.g., translation, in-plane and in-depth rotation, scale) as shown in (Azulay & Weiss, 2019; Engstrom et al., 2019; Fawzi & Frossard, 2015), as well as noise corruptions (Hendrycks & Dietterich, 2019; Geirhos et al., 2018b), and **c**) are vulnerable to imperceptible adversarial image perturbations (Szegedy et al., 2013; Goodfellow et al., 2014; Nguyen et al., 2015). Majority of these works, however, have used either the ImageNet dataset or its variations, and thus might be biased towards ImageNet characteristics. Utilizing a very challenging dataset that has been proposed recently, known as ObjectNet (Barbu et al., 2019), here we seek to answer how well the state of the art CNNs generalize to real world object recognition scenarios. We also explore the role of spatial context in object recognition and answer whether it is better to use cropped objects (using bounding boxes) or segmented objects to achieve higher accuracy and robustness. Furthermore, we study the relationship between object recognition, scene understanding, and object detection. These are important problems that have been less explored.

Several datasets have been proposed for training and testing object recognition models, and to study their generalization ability (e.g., ImageNet by Deng et al. (2009), Places by Zhou et al. (2017), CIFAR by Krizhevsky et al. (2009), NORB by LeCun et al. (2004), and iLab20M by Borji et al. (2016)). As the most notable one, ImageNet dataset has been very instrumental for gauging the progress in object recognition over the past decade. A large number of studies have tested new ideas by training deep models on ImageNet (from scratch), or by finetuning pre-trained (on ImageNet) classification models on other datasets. With the ImageNet being retired, the state of the object recognition problem remains unclear. Several questions such as out of distribution generalization, "superhuman performance" (He et al., 2016) and invariance to transformations persist. To rekindle the discourse, recently Barbu et al. (2019) introduced the ObjectNet dataset which according to their claim has less bias than other recognition datasets[4]. This dataset is supposed to be used solely as a test set and comes with a licence that disallows the researchers to finetune models on it. Images are pictured by Mechanical Turk workers using a mobile app in a variety of backgrounds, rotations, and imaging viewpoints. ObjectNet contains 50,000 images across 313 categories, out of which 113 are in common with ImageNet categories. Astonishingly, Barbu et al. found that the state of the art object recognition models perform drastically lower on ObjectNet compared to their performance on ImageNet (about 40-45% drop). Our principal goal here it to revisit the Barbu et al.'s analysis and measure the actual performance drop on ObjectNet compared to ImageNet. To this end, we limit our analysis to the 113 overlapped categories between the two datasets. We first annotate the objects in the ObjectNet scenes by drawing boxes around them. We then apply a number of deep models on these object boxes and find that models perform significantly better now, compared to their performance on the entire scene (as is done in Barbu et. al). Interestingly, and perhaps against the common belief, we also find that training and testing models on segmented objects, rather than the object bounding box or the full image, leads to consistent improvement in accuracy and robustness over a range of classification tasks and image transformations (geometric, natural distortions, and adversarial attacks). Lastly, we provide a qualitative (and somewhat anecdotal) analysis of extreme cases in object recognition for humans and machines.

## 2 RELATED WORK

**Robustness against synthetic distribution shifts.** Most research on assessing model robustness has been focused on synthetic image perturbations (e.g., spatial transformations, noise corruptions, simulated weather artifacts, temporal changes (Gu et al., 2019), and adversarial examples) perhaps because it is easy to precisely define, implement, and apply them to arbitrary images. While models have improved significantly in robustness to these distribution shifts (e.g., Zhang (2019); Zhang et al. (2019); Cohen & Welling (2016)), they are still not as robust as humans. Geirhos et al. (2018b) showed that humans are more tolerant against image manipulations like contrast reduction, additive noise, or novel eidolon-distortions than models. Further, humans and models behave differently (witnessed by different error patterns) as the signal gets weaker. Zhu et al. (2016) contrast the influence of the foreground object and image background on the performance of humans and models.

**Robustness against natural distribution shifts.** Robustness on real data is a clear challenge for deep neural networks. Unlike synthetic distribution shifts, it is difficult to define distribution shifts that occur naturally in the real-world (such as subtle changes in scene composition, object types, and lighting conditions). Recht et al. (2019) closely followed the original ImageNet creation process

---

[4]ObjectNet dataset, however, has it own biases. It consists of indoor objects that are available to many people, are mobile, are not too large, too small, fragile or dangerous.

**Figure 1:** Sample images from the ObjectNet dataset along with our manually annotated object bounding boxes from `Chairs`, `Teapots` and `T-shirts` categories. The leftmost column shows three chair examples from the ImageNet dataset. ImageNet scenes often have a single isolated object in them whereas images in the ObjectNet dataset contain multiple objects. Further, ObjectNet objects cover a wider range of variation in contrast, rotation, scale, and occlusion compared to ImageNet objects (See arguments in Barbu et al. (2019)). In total, we annotated 18,574 images across 113 categories in common between the two datasets. This figure is modified from Figure 2 in Barbu et al. (2019).

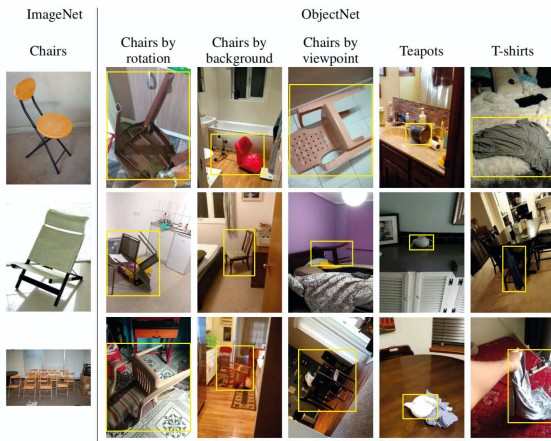

to build a new test set called ImageNetV2. They reported a performance gap of about 11% (top-1 acc.) between the performance of the best deep models on this dataset and the original test set. Similar observations have been made by Shankar et al. (2020). By evaluating 204 ImageNet models in 213 different test conditions, Taori et al. (2020) found that a) current synthetic robustness does not imply natural robustness. In other words, robustness measures for synthetic distribution shifts are weakly predictive of robustness on the natural distribution shifts, b) robustness measurements should control for accuracy since higher robustness can sometimes be explained by the higher accuracy on a standard unperturbed test set, and c) training models on larger and more diverse data improves robustness but does not lead to full closure of the performance gap. A comprehensive benchmark of distribution shifts in the wild, known as WILDS, has recently been published by Koh et al. (2020), encompassing different data modalities including vision. In D'Amour et al. (2020), authors regard "underspecification" a major challenge to the credibility and generalization of modern machine learning pipelines. An ML pipeline is underspecified when it returns models that perform very well on held-out test sets during training but perform poorly at deployment time.

**Contextual interference.** Context plays a significant role in pattern recognition and visual reasoning (e.g., Bar (2004); Torralba & Sinha (2001); Rabinovich et al. (2007); Heitz & Koller (2008); Galleguillos & Belongie (2010)). The extent to which visual context is being used by deep models is still unclear. Unlike models, humans are very good at exploiting context when it is helpful and discard it when it causes ambiguity. In other words, deep models do not understand what is the foreground object and what constitutes the background[5]. Nagarajan et al. (2020) mention that ML models utilize features (e.g., image background) which are spuriously correlated with the label during training. This makes them fragile at the test time when statistics slightly differ. As we argue here, this is one of the main reasons why deep models are so vulnerable to geometric and adversarial perturbations. Geirhos et al. (2020) have studied this phenomenon under the "shortcut learning" terminology from a broader perspective.

**Insights from human vision.** CNNs turn out to be good models of human vision and can explain the first feed-forward sweep of information (See Kriegeskorte (2015) for a review). They, however, differ from human visual processing in several important ways. Current object recognition methods do not rely on segmentation, whereas figure-ground segmentation plays a significant role in human vision, in particular for the encoding of spatial relations between 3D object parts (Biederman, 1987; Serre, 2019). Some computer vision works, predating deep learning, have also shown that pre-segmenting the image before applying the recognition algorithms, improves the accuracy (Malisiewicz & Efros, 2007; Rabinovich et al., 2007; Rosenfeld & Weinshall, 2011). Unlike the human vision system, CNNs are hindered drastically in crowded scenes (e.g., Volokitin et al. (2017)). CNNs rely more on texture whereas humans pay more attention to shape (Geirhos et al., 2018a). Utilizing minimal recognizable images, Ullman et al. (2016) argued that the human visual system uses features and processes that are not used by current deep models.

---

[5]As an example, consider a model that is trained to classify camels *vs.* cows, with camels always shown in sandy backgrounds and cows shown against grassy backgrounds. Although such a model does well during training, it gets confused when presented with cows in sandy backgrounds at test time (Beery et al., 2018). See also Rosenfeld et al. (2018) for another example in the context of object detection

**Figure 2:** Performance of deep object recognition models on ObjectNet dataset. Results of our analysis by applying AlexNet, VGG-19, and ResNet-152 models to object bounding boxes (i.e., isolated objects) are shown in blue (overlaid in Fig. 1 from the ObjectNet paper). Feeding object boxes to models instead of the entire scene improves the accuracy about 10-15%. The right panel shows the results using our code (thus leading to a fair comparison). Performance improvement is more significant now which is ∼ 20-30%. We did not use test time data augmentation here.

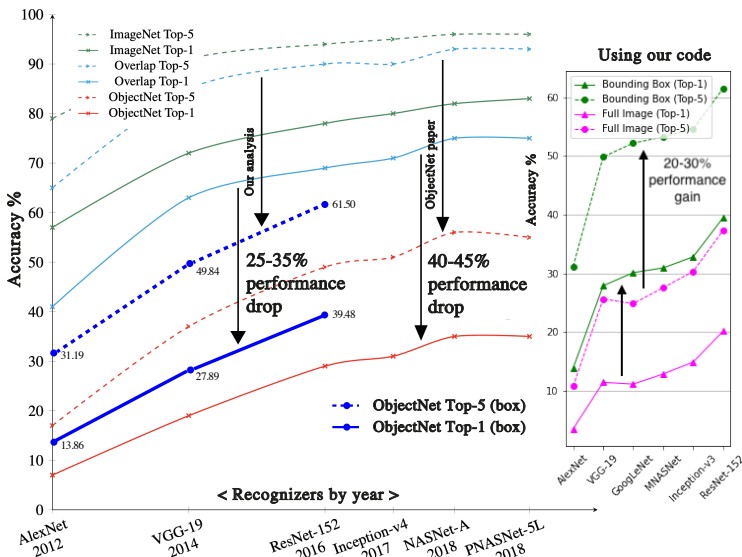

## 3 EXPERIMENTS AND RESULTS

### 3.1 ACCURACY AND ROBUSTNESS AGAINST NATURAL DISTRIBUTION SHIFTS

**A critic of Barbu et al. (2019).** Barbu et al.'s work is a great contribution to the field to answer how well object recognition models generalize to the real-world circumstances and to control for biases in data collection. It, however, suffers from a major shortcoming that is making no distinction between "object detection" and "object recognition". This confusion brings along several concerns:

1. They use the term "object detector" to refer to "object recognition" models. Object detection and object recognition are two distinct, yet related, tasks. Each one has its own models, datasets, evaluation measures, and inductive biases. For example, as shown in Fig. 1, images in object recognition datasets (e.g., ImageNet) often contain a single object, usually from a closeup view, whereas scenes in object detection datasets (e.g., MS COCO (Lin et al., 2014), OpenImages (Kuznetsova et al., 2018)) usually have multiple objects. Objects in the detection datasets vary more in some parameters such as occlusion and size. For instance, there is a larger variation in object scale in detection datasets (Singh & Davis, 2018). This discussion also relates to the distinction between "scene understanding" and "object recognition". To understand a complex scene, as humans we look around, fixate on individual objects to recognize them, and accumulate information over fixations to perform more complex tasks such as answering a question or describing an event. To avoid biases in recognition datasets (e.g., typical scales or object views), we propose to (additionally) use detection datasets to study object recognition. We will discuss this further in Section 4.

2. Instead of applying models to isolated objects, Barbu et al. apply them to cluttered scenes containing multiple objects. Unlike ImageNet where the majority of images include only a single object, ObjectNet images have multiple objects in them and are often more cluttered. Therefore, the drop in performance of models on ObjectNet can be merely due to the fact that pretrained models on ImageNet have been trained on individual objects.

3. In addition to top-1 accuracy, Barbu et al. also report top-5 accuracy. One might argue that this may suffice in dealing with scenes containing multiple objects. Top-5 accuracy was first introduced in Russakovsky et al. (2015) to remedy the issues with the top-1 accuracy. The latter can be overly stringent by penalizing predictions that appear in the image but do not correspond to the target label. Top-5 accuracy itself, however, has two shortcomings. First, a model can still be penalized if all of the five guesses exist in the image, but none is the image label. Both scores fall short in addressing the images with counter-intuitive labels (e.g., when non-salient objects are labeled; Appx. E). Second, on fine-grained classification tasks (ImageNet has several fine-grained classes e.g., dogs), allowing five

predictions can make certain class distinctions trivial (Shankar et al., 2020). For example, there are five turtles in the ImageNet class hierarchy (mud turtle, box turtle, loggerhead turtle, leatherback turtle, and terrapin) that are difficult to distinguish. A classifier may trick the score by generating all of these labels for a turtle image to ensure it predicts the correct label. Shankar et al. proposed to use multi-label accuracy as an alternative to top-5 score. Each image has a set of target labels (i.e., multi-label annotations). A prediction is marked correct if it corresponds to any of the target labels for that image. This score, however, may favor a model that generates correct labels but may confuse the locations over a model that is spatially more precise but misses some objects (See also Beyer et al. (2020)). Regardless, since multi-label annotations for ObjectNet are not available, we report both top-1 and top-5 scores when feeding isolated objects to models.

**Bounding box annotation.** The 113 object categories in the ObjectNet dataset, overlapped with the ImageNet, contain 18,574 images in total. On this subset, the average number of images per category is 164.4 (*min*=55, *max*=284). Fig. 8 in Appx. A shows the distribution of the number of images per category on this dataset (*envelope* and *dish drying rack* are the most and least frequent objects, respectively). We drew a bounding box around the object corresponding to the category label of each image. If there were multiple nearby objects from the same category (e.g., chairs around a table), we tried to include all of them in the bounding box. Some example scenes and their corresponding bounding boxes are given in Fig. 1. Appx. H shows more stats on ObjectNet.

**Object recognition results.** We employ six widely-used state of the art deep neural networks including AlexNet (Krizhevsky et al., 2012), VGG-19 (Simonyan & Zisserman, 2014), GoogLeNet (Szegedy et al., 2015), ResNet-152 (He et al., 2016), Inception-v3 (Szegedy et al., 2016)[6], and MNASNet (Tan et al., 2019). AlexNet, VGG-19, and ResNet-152 have also been used in the ObjectNet paper (Barbu et al., 2019). We use the PyTorch implementation of these models[7]. Since the code from the ObjectNet paper is unavailable (at the time of preparing this work), in addition to applying models to bounding boxes and plotting the results on top of the results from the ObjectNet paper, we also run our code on both the bounding boxes and the full images. This allows a fair comparison and helps mitigate possible inconsistency in data processing methods (e.g., different data normalization schemes or test time data augmentation such as rotation, scale, color jittering, cropping, etc.).

Fig. 2 shows an overlay of our results in Fig. 1 from the ObjectNet paper. As can be seen, applying models to the object bounding box instead of the entire scene improves the accuracy about 10-15%. Although the gap is narrower now, models still significantly underperform on ObjectNet than the ImageNet dataset. Using our code, the improvement going from full image to bounding boxes is around 20-30% across all tested models (the right panel in Fig. 2). Our results using the full image are lower than Barbu et al.'s results using the full image (possibly because we do not utilize data augmentation). This relative difference entails that applying their code to bounding boxes will likely improve the performance beyond 10% that we obtained here. Assuming 25% gain in performance on top of their best results when using boxes, will still not close the performance gap which indicates that ObjectNet remains a challenging dataset for testing object recognition models.

Breakdown of accuracy over the 113 categories is shown in Appx. B (Figs. 9 & 10 over isolated objects and Figs. 11 & 12 over the full image). Interestingly, in both cases, almost all models, except GoogLeNet on isolated objects and AlexNet on the full image, perform the best over the *safety pin* category. Inspecting the images from this class, we found that they have a single safety pin often held by a person (perhaps about the same distance from the camera thus similar scales). The same story is true about the *banana* class which is the second easiest category using the bounding boxes. This object becomes much harder to recognize when using the full image (26.88% *vs.* 70.3% using boxes) which highlights the benefit of applying models to isolated objects rather than scenes.

## 3.2 ACCURACY AND ROBUSTNESS AGAINST SYNTHETIC DISTRIBUTION SHIFTS

### 3.2.1 ROBUSTNESS AGAINST COMMON IMAGE CORRUPTIONS

Previous work has shown that ImageNet-trained CNNs generalize poorly over a wide range of image distortions (e.g., Hendrycks & Dietterich (2019); Azulay & Weiss (2019); Dodge & Karam (2017)). These works, however, have applied CNNs to the whole scene. Here, we

---

[6]Barbu et al. have used Inception-v4.
[7]https://pytorch.org/docs/stable/torchvision/models.html

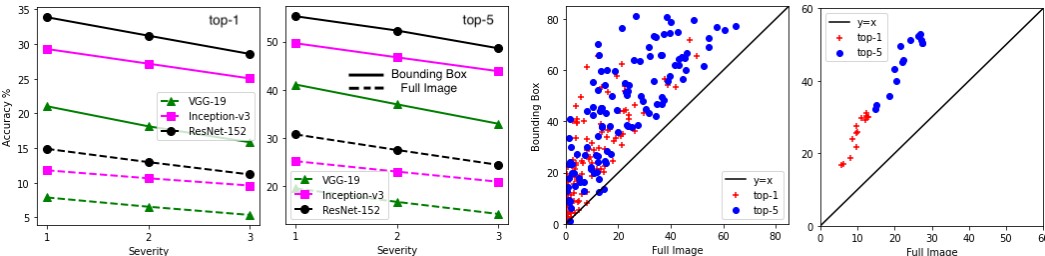

**Figure 3:** Left two panels) Average top-1 and top-5 accuracy of models over 1130 images from the ObjectNet dataset corrupted by 14 natural image distortions. Right two panels) Average classification accuracy per each of the 113 categories (left) and each of the 14 distortion types (right). In all cases, applying models to the isolated object (using bounding boxes) leads to higher robustness than applying them to the full image.

ask whether applying the models to the bounding boxes can improve robustness against image distortions. Following Hendrycks & Dietterich (2019), we systematically test how model accuracy degrades if images are corrupted by 14 different types of distortions including `Gaussian noise`, `shot noise`, `impulse noise`, `defocus blur`, `glass blur`, `motion blur`, `zoom blur`, `snow`, `frost`, `fog`, `brightness`, `contrast`, `elastic transform`, and `JPEG compression` at 3 levels of corruption severity. Fig. 36 (Appx. F) shows sample images along with their distortions. Ten images from each of the 113 categories of ObjectNet (1130 images in total) were fed to three models including VGG-19, Inception-v3, and ResNet-152.

Aggregate results over the full image and the object bounding box (both resized to 224 × 224 pixels) are shown in Fig. 3. All three models are more robust when applied to the object bounding box than the full image at all corruption levels, using both top-1 and top-5 scores (left two panels). Among models, ResNet-152 performs better and is the most robust model. It is followed by the Inception-v3 model. For nearly all of the 113 object categories, using bounding boxes leads to higher robustness than using the full image (the third panel). Similarly, using bounding boxes results in higher robustness against all distortion types (the right-most panel). Across distortion types, shown in Figs. 37 & 38 (Appx. F), ResNet-152 consistently outperforms the other two models at all severity levels, followed by Inception-v3. It seems that models are hindered more by *impulse noise*, *frost*, *zoom blur*, and *snow* distortions. The top-1 accuracy at severity level 2 on these distortions is below 20%. Overall, we conclude that limiting the object area only to the bounding box leads not only to higher prediction accuracy but also to higher robustness against image distortions. Extrapolating this approach, can we improve robustness by shrinking the object region even further by using the segmentation masks? We will thoroughly investigate this question in the next subsections.

### 3.2.2 ROBUSTNESS AGAINST ADVERSARIAL PERTURBATIONS

Despite being very accurate, CNNs are highly vulnerable to adversarial inputs (Szegedy et al., 2013; Goodfellow et al., 2014). These inputs are crafted carefully and maliciously by adding small imperceptible perturbations to them (e.g., altering the value of a pixel up to 8 units under the $\ell_\infty$-norm; pixels in the range [0, 255]). Here we apply the ImageNet pretrained models to 1130 images that were selected above. The models are tested against the Fast Gradient Sign Method (FGSM) (Goodfellow et al., 2014) at two perturbation budgets in the untargeted white-box setting.

Table. 1 shows the results. We find that models are more resilient against the FGSM attack when applied to the bounding box than the full image. While the input size is the same in both cases (224×224), the adversary has more opportunity to mislead the

|  | Full Image | | Bounding Box | |
| Model | $\epsilon = 2/255$ | $\epsilon = 8/255$ | $\epsilon = 2/255$ | $\epsilon = 8/255$ |
| --- | --- | --- | --- | --- |
| VGG-19 | 0.53/2.48 | 0.09/0.71 | 3.27/10.44 | 1.06/5.66 |
| Inception-v3 | **3.18/10.62** | **1.42**/4.42 | 9.03/25.13 | 4.87/15.6 |
| ResNet-152 | 2.39/10.34 | 1.15/**4.96** | **10.62/26.64** | **6.64/19.73** |

**Table 1:** Robust accuracy (top-1/top-5) against FGSM attack.

classifier in the full image case since a larger fraction of pixels play an insignificant role in the decisions made by the network. This aligns with observations from the visualization tools (e.g., Selvaraju et al. (2017)) revealing that CNNs indeed rely only on a small subset of image pixels to elicit a decision. One might argue that the lower robustness on the full images could be due to training and test discrepancy (i.e., training models on single objects and applying them to the entire scene). To address this, in the next subsection we train and test models in the same condition.

## 3.3 THE INFLUENCE OF THE SURROUNDING CONTEXT ON ROBUSTNESS

Despite a large body of literature on whether and how much visual context benefits CNNs in terms of accuracy and robustness[8], the matter has not been settled yet (e.g., Bar (2004); Torralba & Sinha (2001); Rabinovich et al. (2007); Rosenfeld et al. (2018); Heitz & Koller (2008); Divvala et al. (2009); Zhu et al. (2016); Xiao et al. (2020); Malisiewicz & Efros (2007)). To study how context surrounding an object impacts model accuracy and robustness in more detail, we conducted

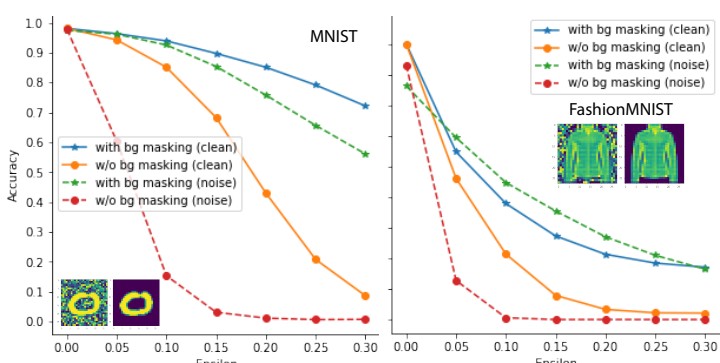

**Figure 4:** The effect of background subtraction (a.k.a foreground detection) on adversarial robustness (here against the FGSM attack). Two models are trained and tested on clean and noisy data from MNIST (left) and FashionMNIST (right) datasets. In the noise case, the object is overlaid in a white noise field (no noise on the object itself).

two experiments. In the first one, we trained two CNNs (2 conv layers, each followed by a pooling layer and 2 final fc layers) on MNIST and Fashion MNIST datasets, for which it is easy to derive the foreground masks (Figs. 39 & 40; Appx. G). CNNs were trained on either the *original clean images* or the *foreground objects placed on a white noise background*. We then tested the models against the FGSM attack w/o background subtraction. With background subtraction, we essentially assume that the adversary has access only to the foreground object (i.e., effectively removing the perturbations that fall on the background). As results in Fig. 4 show, background subtraction improves the robustness substantially using both models and over both datasets.

To examine whether the above conclusion generalizes to more complex natural scenes, we ran a second experiment. First, we selected images from ten classes of the MS COCO dataset including `chair`, `car`, `book`, `bottle`, `dinning table`, `umbrella`, `boat`, `motorcycle`, `sheep`, and `cow`. Objects from these classes come with a segmentation mask (one object per image; 100 images per category; 1000 images in total). Around 32.7% of the image pixels fall inside the object bounding box and around 58.1% of the bounding box pixels fall inside the object mask. Fig. 5 shows a sample chair alongside its bounding box and its segmentation mask.

We then trained three ResNet-18 models (finetuned on ImageNet), one per each input type: 1) *full image*, 2) *bounding box*, and 3) *segmented object* (placed in a dark background). Models were trained on 70 images per category (700 in total) for 10 epochs and were then tested on the remaining 30 images per category. An attempt was made to tune the parameters to attain the best test accuracy in each case (e.g., by avoiding overfitting). The test accuracy[9] of models in order are 66.9%, 78%, and 80.3%. One reason

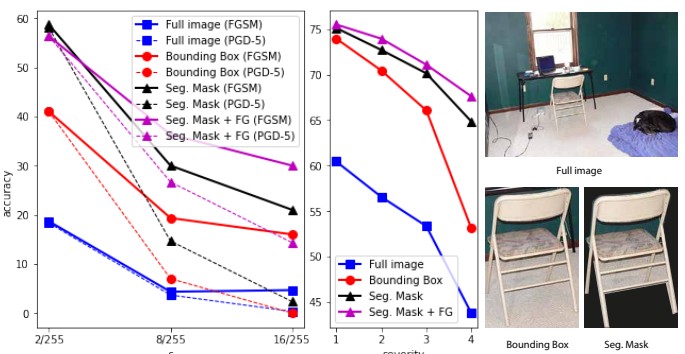

**Figure 5:** Model accuracy against adversarial perturbations (left) and noise corruptions (middle). The right panel shows a sample chair image along with its bounding box and segmentation mask.

behind lower prediction accuracy using boxes might be because multiple objects may fit inside

---

[8]Majority of such works are focused on model accuracy

[9] Taori et al. (2020) argue that robustness scores should control for accuracy as more predictive models in general are more robust. To avoid this issue we used models that have about the same standard accuracy.

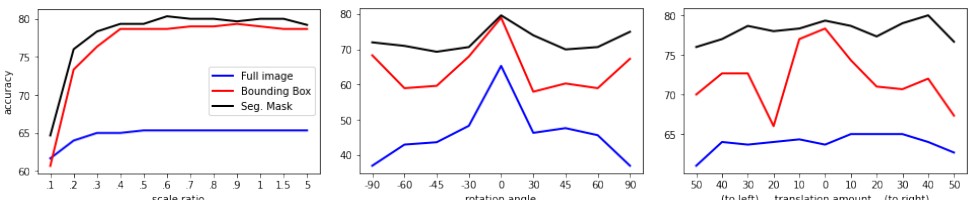

**Figure 6:** Performance (top-1) of the ResNet-18 model against geometric transformations.

the bounding box (e.g., for elongated objects such as *broom*). Model performance against FGSM and $\ell_\infty$ PGD-5 (Projected Gradient Descent by Madry et al. (2017)) adversarial attacks are shown in Fig. 5 (left panel). We observe that training models on segmented objects leads to higher adversarial robustness against both types of attacks. The improvement is more pronounced at higher perturbations. We also considered a condition in which we masked the perturbations that fall on the background, denoted as "Seg. Mask + FG" in the figure. We noticed even higher robustness against the attacks by removing the background perturbations. These results encourage using foreground detection as an effective adversarial defense.

The middle panel in Fig. 5 shows model robustness against noise corruptions (averaged over the 14 distortions used in Section 3.2.1). Here again, we find that using segmentation masks leads to higher robustness compared to the full image and object boxes. "Seg. Mask + FG" leads to the best robustness among the input types. While it might be hard to draw a general conclusion regarding the superiority of the segmentation masks over bounding boxes in object recognition accuracy, our investigation suggests that using masks leads to a significant boost in adversarial robustness with little or no drop in standard accuracy. Our results offer an upper bound in the utility of segmentation masks in robustness. More work is needed to incorporate this feat in CNNs (i.e., using attention).

### 3.3.1 ROBUSTNESS AGAINST GEOMETRIC TRANSFORMATIONS

We also tested the ResNet-18 model (i.e., trained over the full image, the bounding box, and the segmented object on ObjectNet; as above) against three geometric transformations including *scaling*, *in-plane rotation*, and *horizontal translation*. Fig. 6 shows the results over the 300 test images that were used in the previous subsection. We find that the model trained on segmentation masks is more robust than the other two models over all three geometric transformations, followed by the models trained on the object bounding boxes and the full image, in order.

### 3.4 QUALITATIVE INSPECTION OF OBJECTNET IMAGES AND ANNOTATIONS

During the annotation of ObjectNet images, we came across the following observations: **a)** Some objects look very different when they are in motion (e.g., the fan in row 4 of Fig. 34 in Appx. D), or when they are shadowed or occluded by other objects (e.g., the hammer in Fig. 34 row 4), **b)** Some object instances differ a lot from the typical instances in the same class (e.g., the helmet in Fig. 34 row 5; the orange in Fig. 33 row 5), **c)** Some objects can be recognized only through reading their captions (e.g., the pet food container in Fig. 33 row 2), **d)** Some images have wrong labels (e.g., the pillow in Fig. 33 row 2; the skirt in Fig. 33 row 1; the tray in Fig. 34 row 2; See also Appx. E), **e)** Some objects are extremely difficult for humans (e.g., the tennis racket in Fig. 34 row 4; the shovel in Fig. 33 row 4; the tray in Fig. 33 row 1), **f)** In many images, objects are occluded by hands holding them (e.g., the sock and the shovel in Fig. 33 row 4), **g)** Some objects are hard to recognize in dim light (e.g., the printer in Fig. 33 row 2), and **h)** Some categories are often confused with other categories in the same set. Example sets include {*bath towel, bed sheet, full sized towel, dishrag or hand towel*}, {*sandal, dress shoe (men), running shoe*}, {*t-shirt, dress, sweater, suit jacket, skirt*}, {*ruler, spatula, pen, match*}, {*padlock, combination lock*}, and {*envelope, letter*}.

The left panel in Fig. 7 shows four easy (highly confident correct predictions) and four hard (highly confident misclassifications) for ResNet-152 over six ObjectNet categories. In terms of the difficulty level, easy(difficult) objects for models appear easy(difficult) to humans too. Also, our qualitative inspection shows that ObjectNet includes a large number of objects that can be recognized only after a careful examination (the right panel in Fig. 7). More examples are given in Appx. C.

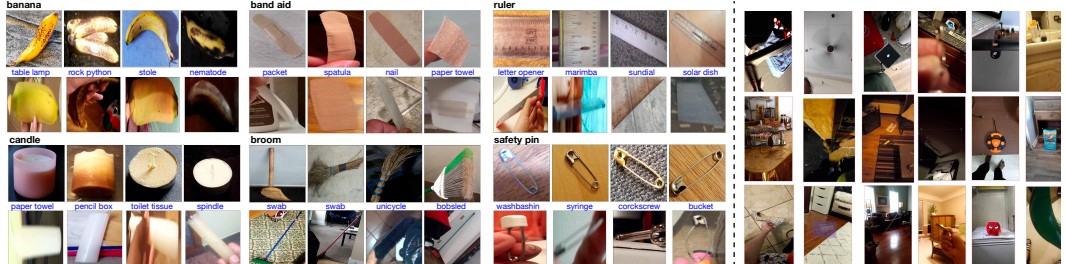

**Figure 7:** Left) Four easiest (highest classification confidence; 1st rows) and four hardest (lowest confidence; 2nd rows) samples per category from the ObjectNet dataset for the ResNet-152 model (predictions shown in blue). The easy samples seem to be easier for humans (qualitatively), while the harder ones also seem to be harder. Right) A collection of challenging objects from the ObjectNet dataset for humans. Can you guess the category of the annotated objects in these images? Keys are: `row 1`: *skirt, fan, desk lamp, safety pin, still camera, and spatula*, `row 2`: *vase, shovel, vacuum cleaner, printer, remote control, and pet food container*, and `row 3`: *sandal, vase, match, spatula, stuffed animal, and shovel*. Appxs. C & D contain more examples.

## 4 TAKEAWAYS AND DISCUSSION

Our investigation reveals that deep models perform significantly better when applied to isolated objects rather than the entire scene. The reason behind this is two-fold. First, there is less variability in single objects compared to scenes containing multiple objects. Second, deep models (used here and also in ObjectNet paper) have been trained on ImageNet images which are less cluttered compared to the ObjectNet images. We anticipate that training models from scratch on large scale datasets that contain isolated objects will likely result in even higher accuracy. Assuming around 30% increase in performance (at best) on top of Barbu et al.'s results using bounding boxes still leaves a large gap of at least 15% between ImageNet and ObjectNet which means that ObjectNet is indeed much harder. It covers a wider range of variations than ImageNet including object instances, viewpoints, rotations, occlusions, etc which pushes the limits of object recognition models. Hence, despite its limitations and biases, ObjectNet dataset remains a great platform to test deep models in realistic situations.

We envision four research directions for the future work in this area. First, background subtraction is a promising mechanism and should be investigated further over large scale datasets (given the availability of high-resolution masks; e.g., MS COCO). We found that it improves robustness substantially over various types of image perturbations and attacks. Humans can discern the foreground object from the image background with high precision. This feat might be the key to robustness and hints towards an interplay and feedback loop between recognition and segmentation that is currently missing in CNNs. Second, measuring human performance on ObjectNet will provide a useful baseline for gauging model performance. Barbu et. al report an accuracy of around 95% when they asked subjects to mention the objects that are present in the scene. This task, however, is different from recognizing isolated objects similar to the regime that was considered here (i.e., akin to rapid scene categorization tasks; See Serre et al. (2007)). Besides, error patterns of models and humans (e.g., Borji & Itti (2014)), in addition to crude accuracy measures, will inform us about the differences in object recognition mechanisms between humans and machines. It could be that models work in a completely different fashion than the human visual system. Third, as discussed in Section 3.1, multi-label prediction accuracy is more appropriate for evaluating recognition models. Annotating all objects in ObjectNet images will thus provide an additional dimension to assess models. In this regard, we propose a new task where the goal is to recognize objects in their natural contexts. This task resembles (cropped) object recognition and object detection, but it is slightly different (i.e., the goal here is to recognize an object limited by a bounding box given all available information in the scene). This is essentially an argument against the recognition-detection dichotomy. Finally, it would be interesting to see how well the state of the art object detectors perform on the ObjectNet dataset (e.g., over overlapped classes between ObjectNet and MS COCO (Lin et al., 2014)). We expect a significant drop in detection performance since it is hard to recognize objects in this dataset.

From a broader perspective, our study reinforces the idea that there is more to scene understanding then merely learning statistical correlations. In particular, background subtraction and visual context are crucial in robust recognition and demand further investigation in future studies.

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

# A  Frequency of the images per category

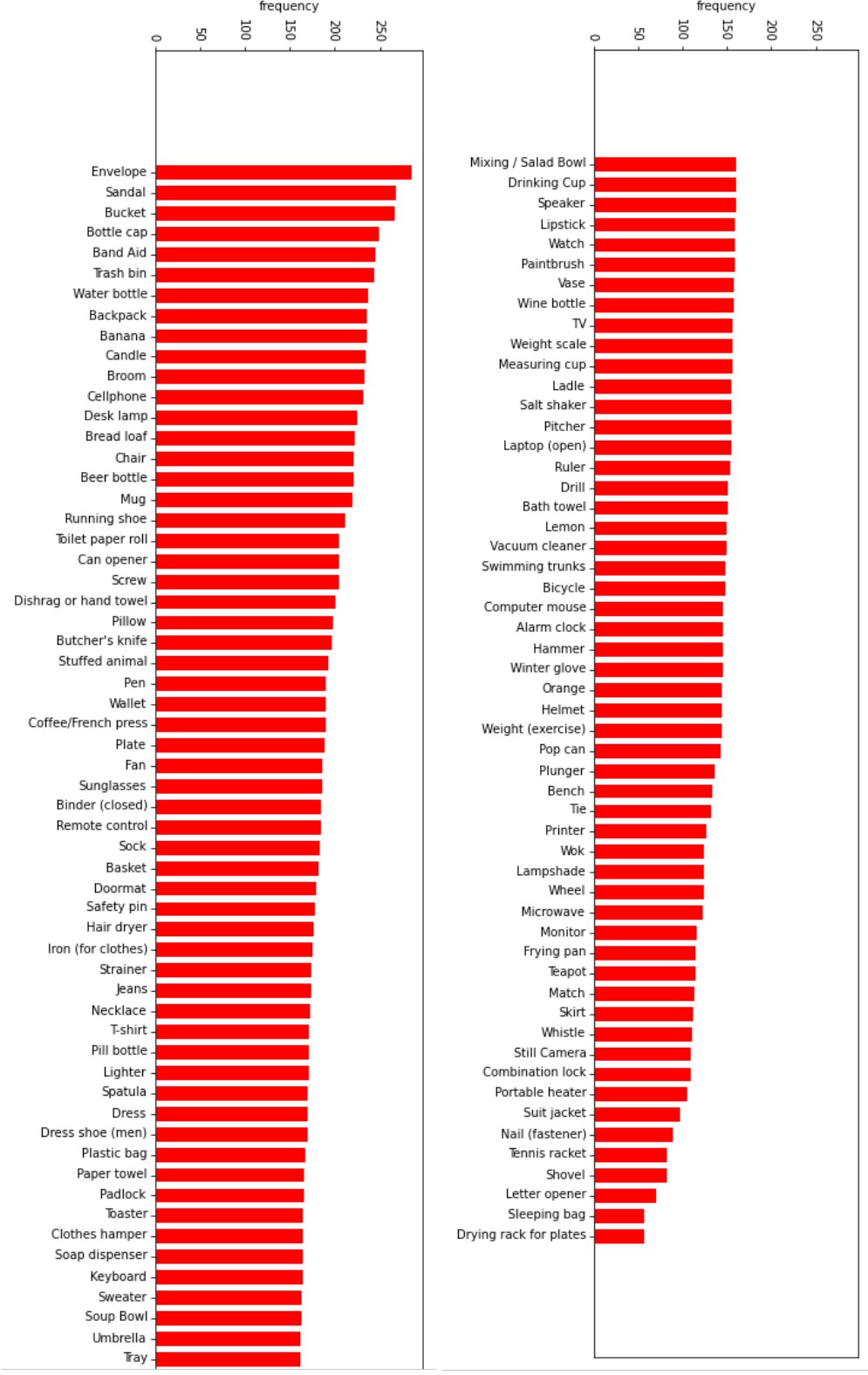

**Figure 8:** Frequency of the images (one object label per image) over the 113 categories of ObjectNet overlapped with the ImageNet. The right bar chart is the continuation of of the left one.

# B  MODEL ACCURACY PER CATEGORY USING BOXES VS. FULL IMAGE

**Figure 9:** Performance of models on object bounding boxes (from left to right: AlexNet, VGG, and GoogLeNet).

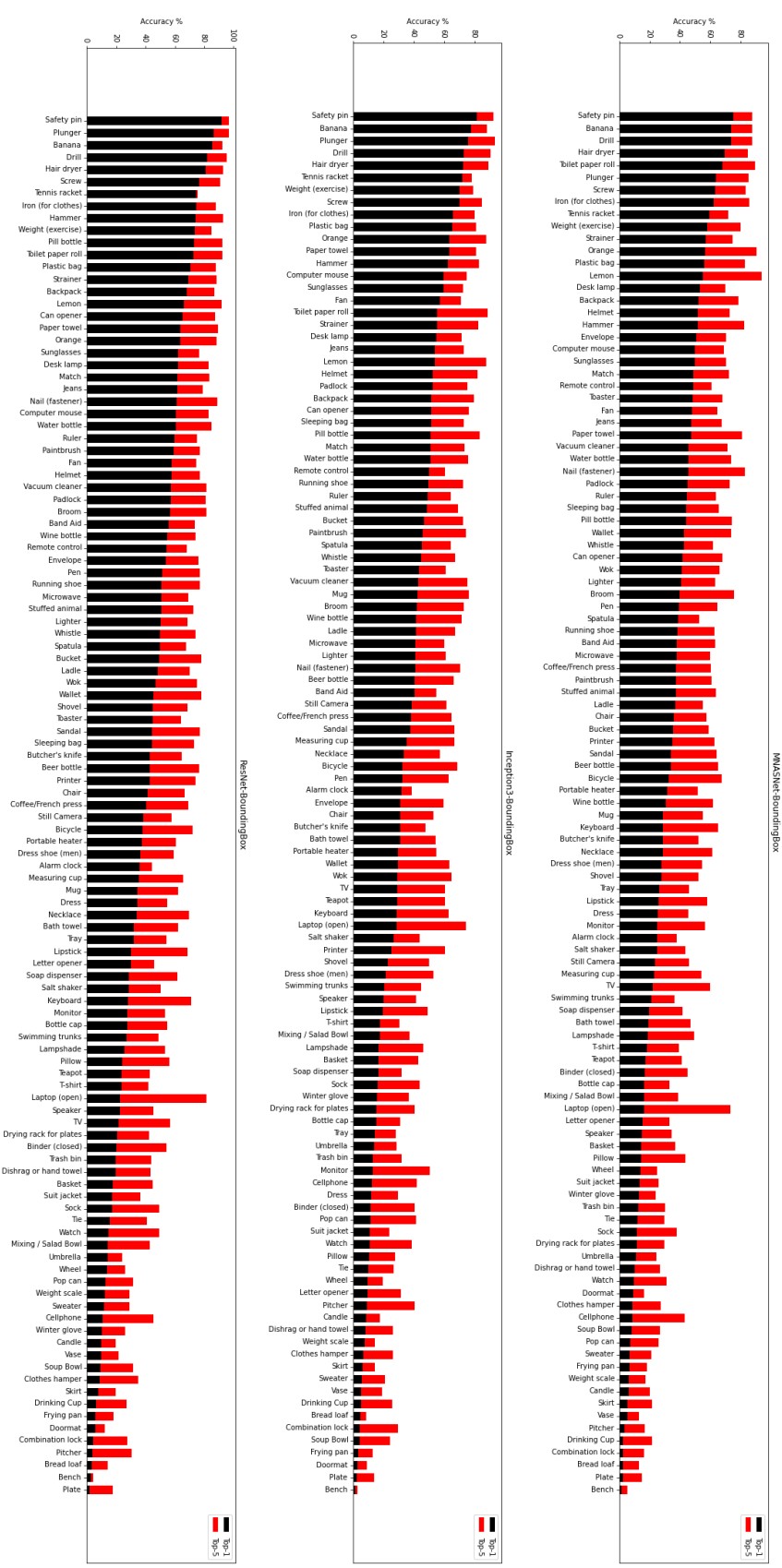

**Figure 10:** Performance of models on object bounding boxes (from left to right: ResNet, Inception3, and MNASNet).

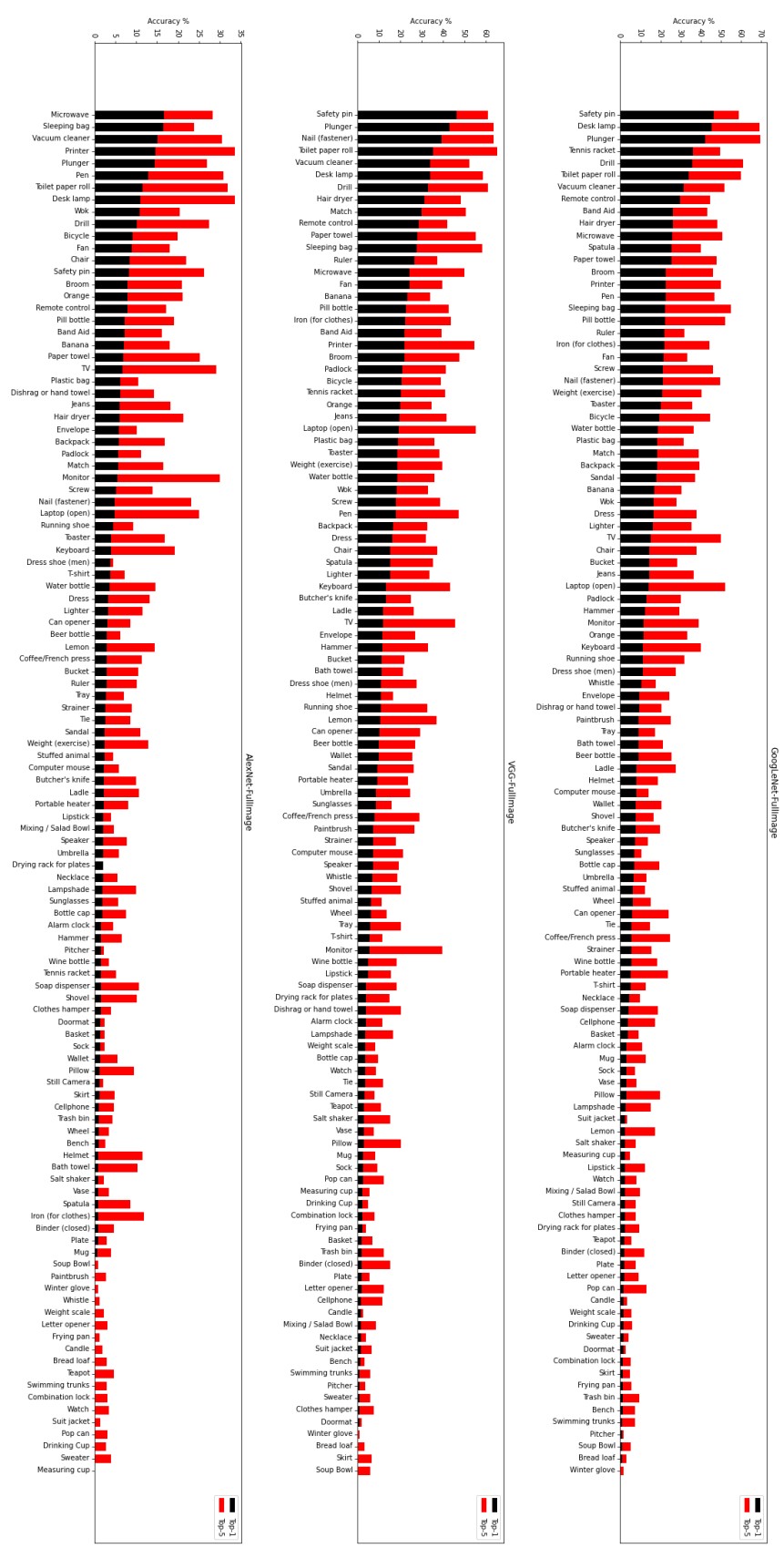

**Figure 11:** Performance of models on the entire image (from left to right: AlexNet, VGG, and GoogLeNet).

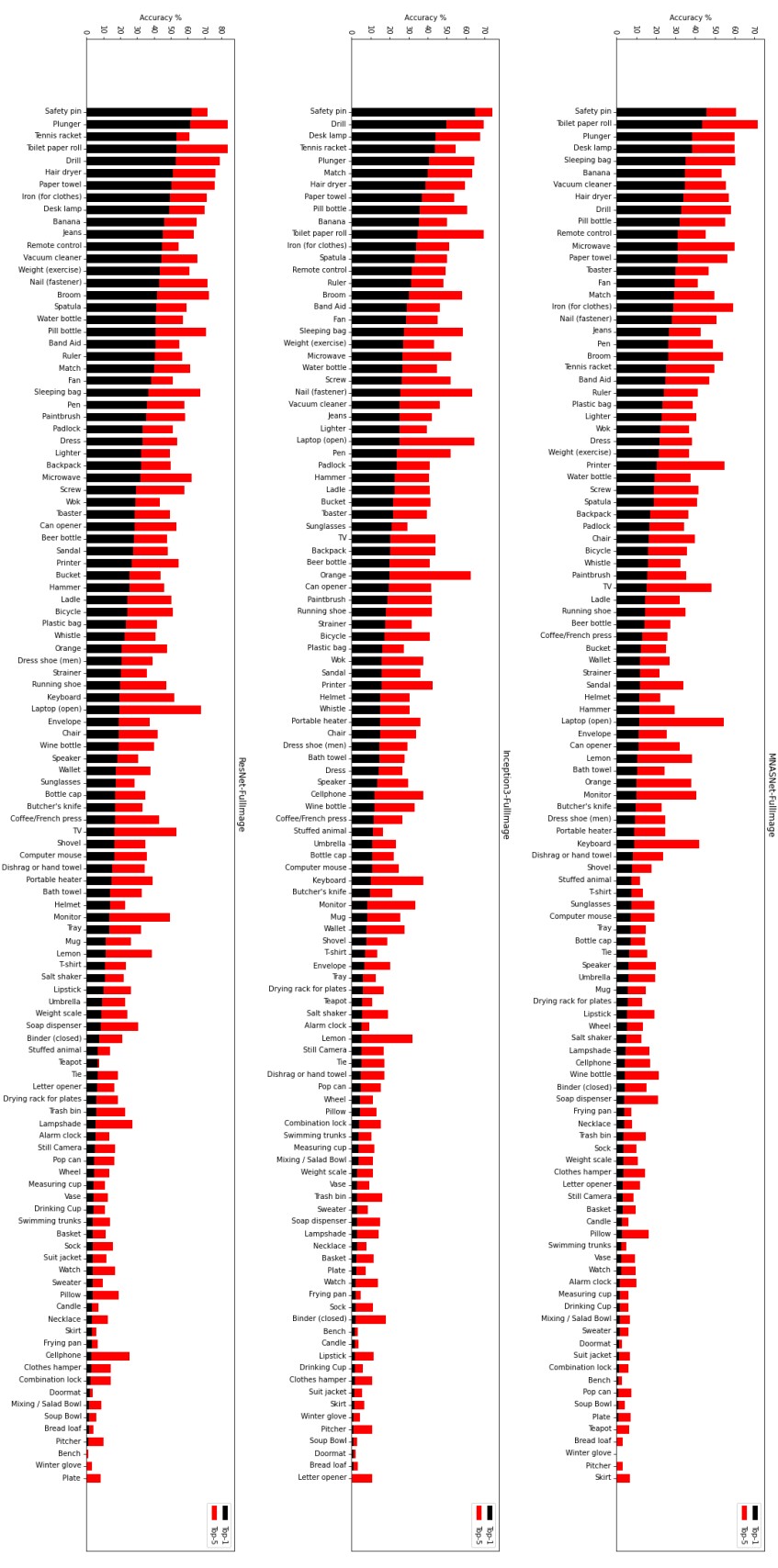

**Figure 12:** Performance of models on the entire image (from left to right: ResNet, Inception3, and MNASNet).

## C EASIEST AND HARDEST OBJECTS FOR THE RESNET-152 MODEL

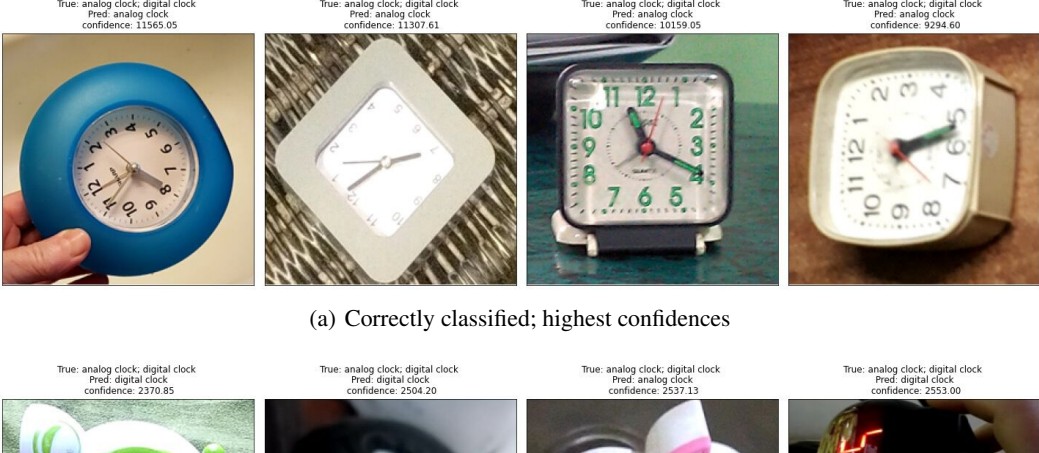

(a) Correctly classified; highest confidences

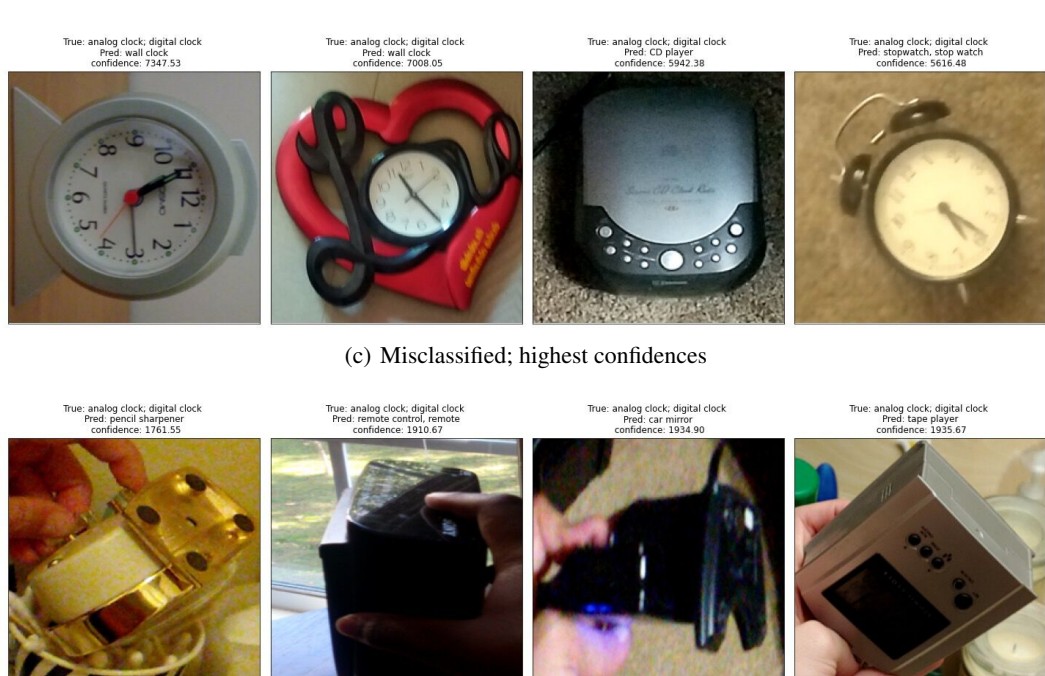

(b) Correctly classified; lowest confidences

(c) Misclassified; highest confidences

(d) Misclassified; lowest confidences

**Figure 13:** Correctly classified and misclassified examples from the Alarm clock class by the ResNet model.

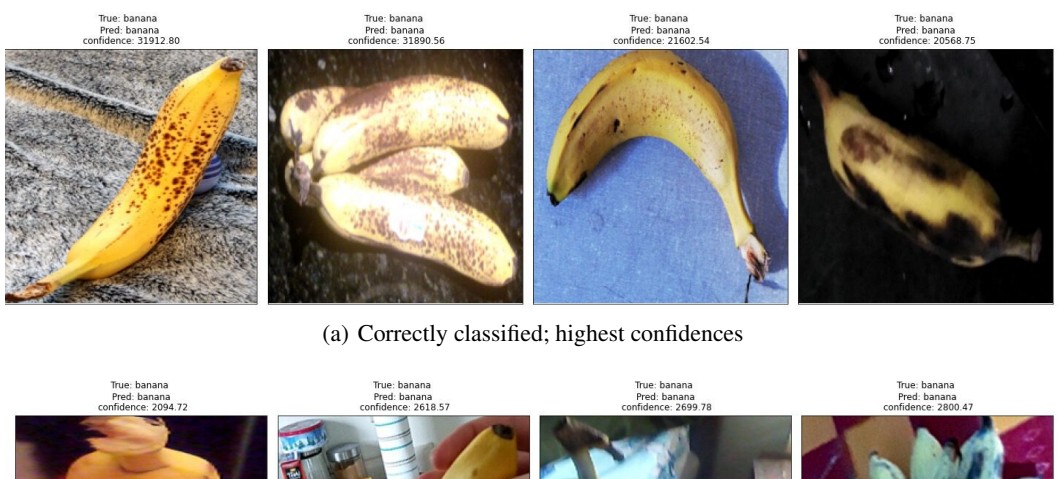

(a) Correctly classified; highest confidences

(b) Correctly classified; lowest confidences

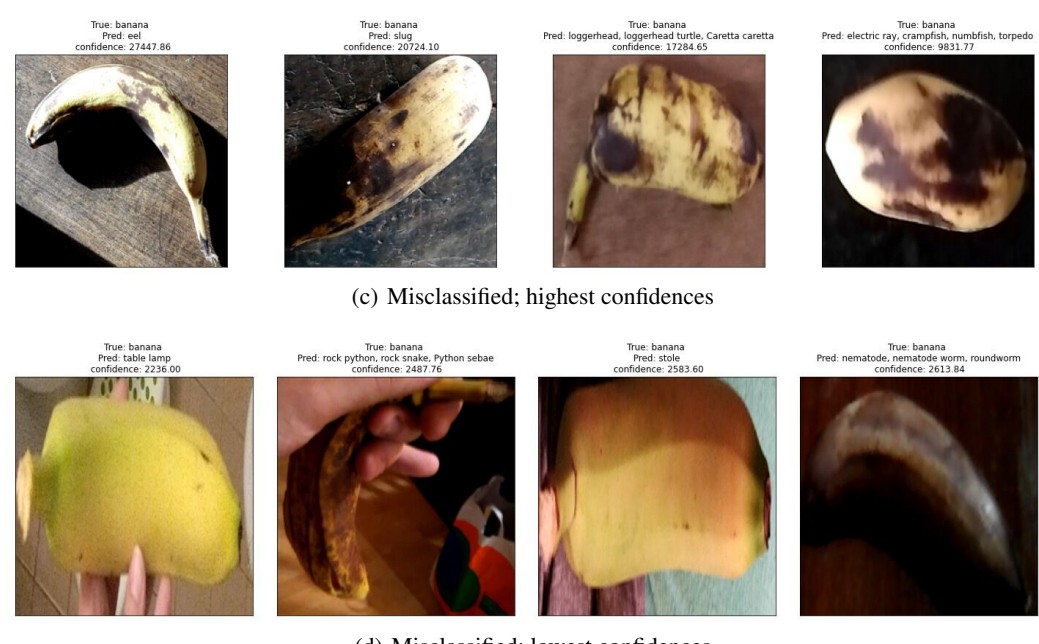

(c) Misclassified; highest confidences

(d) Misclassified; lowest confidences

**Figure 14:** Correctly classified and misclassified examples from the Banana class by the ResNet model.

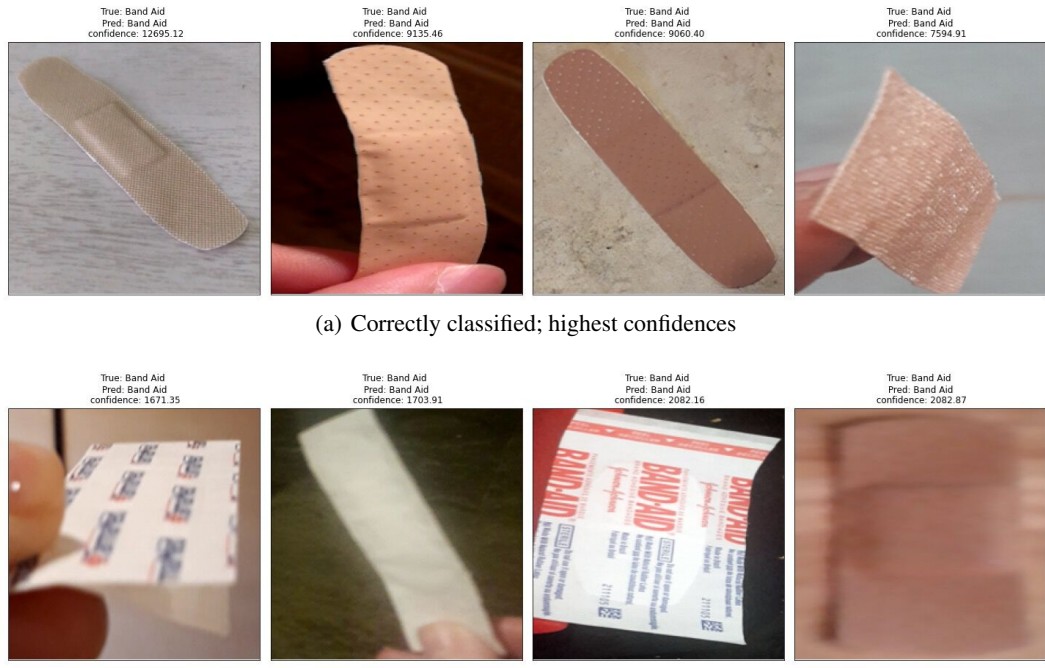

(a) Correctly classified; highest confidences

(b) Correctly classified; lowest confidences

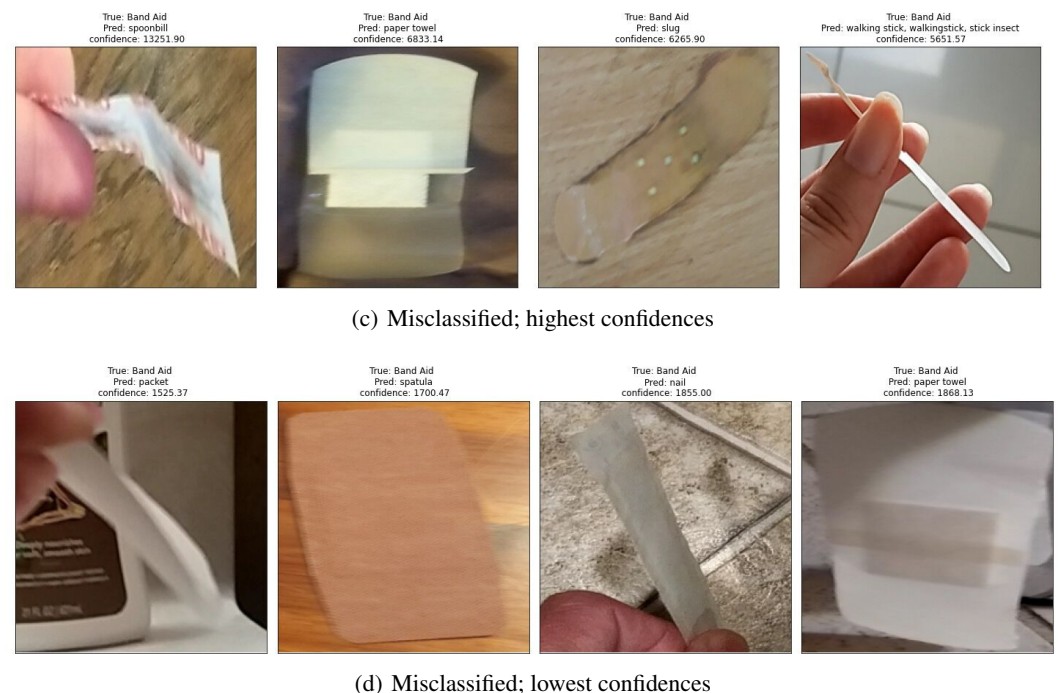

(c) Misclassified; highest confidences

(d) Misclassified; lowest confidences

**Figure 15:** Correctly classified and misclassified examples from the Band-Aid class by the ResNet model.

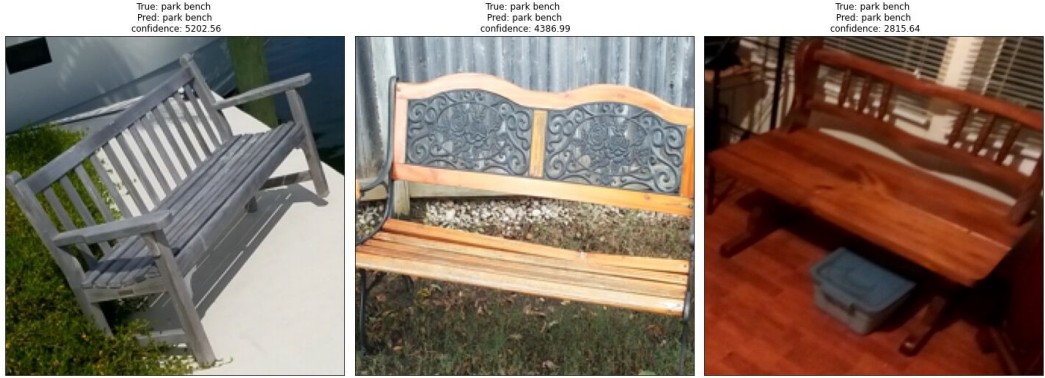

(a) Correctly classified; highest confidences. Only three benches were correctly classified.

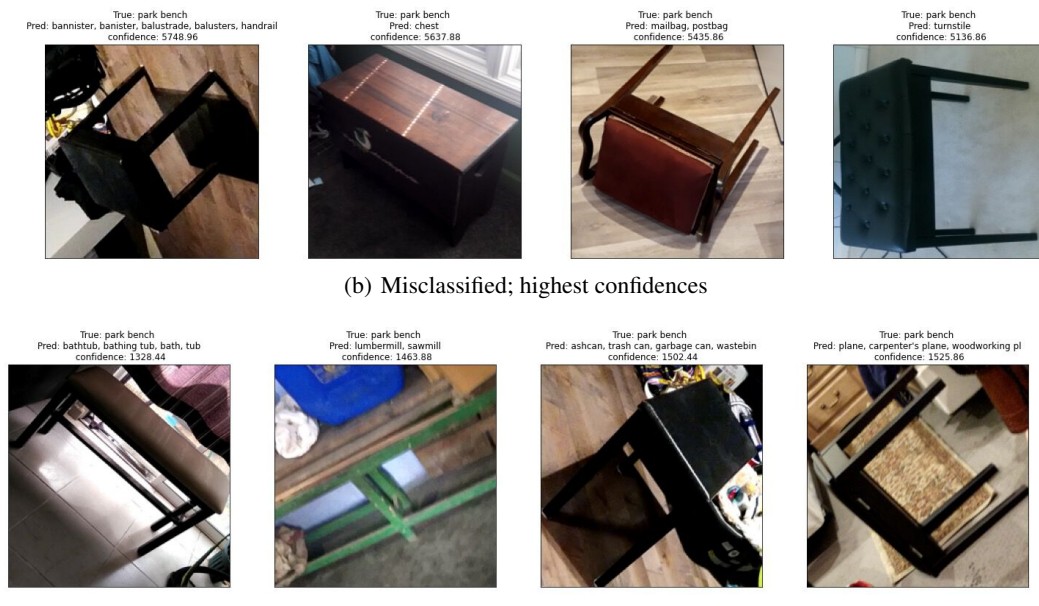

(b) Misclassified; highest confidences

(c) Misclassified; lowest confidences

**Figure 16:** Correctly classified and misclassified examples from the Bench class by the ResNet model.

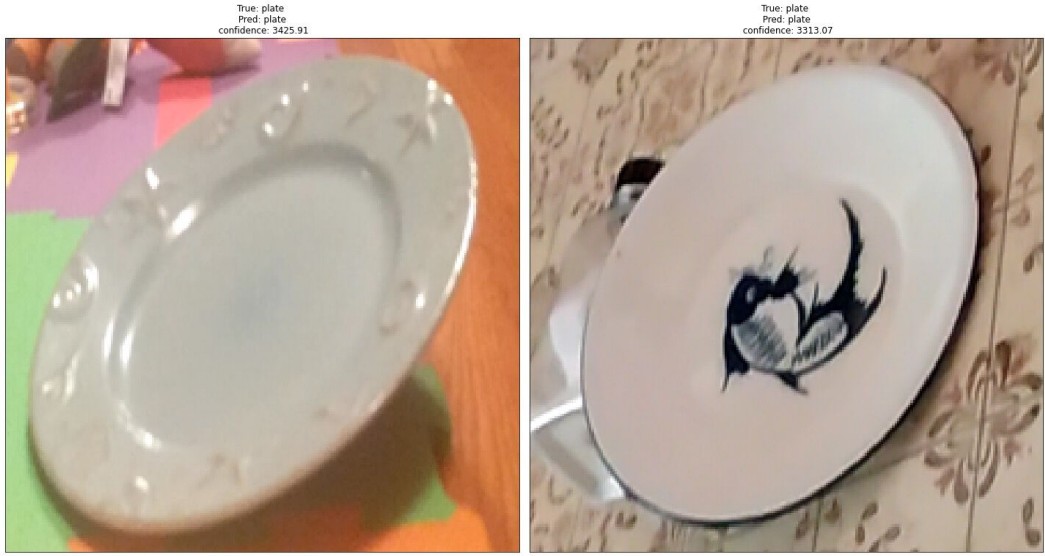

(a) Correctly classified; highest confidences. Only two plates were correctly classified.

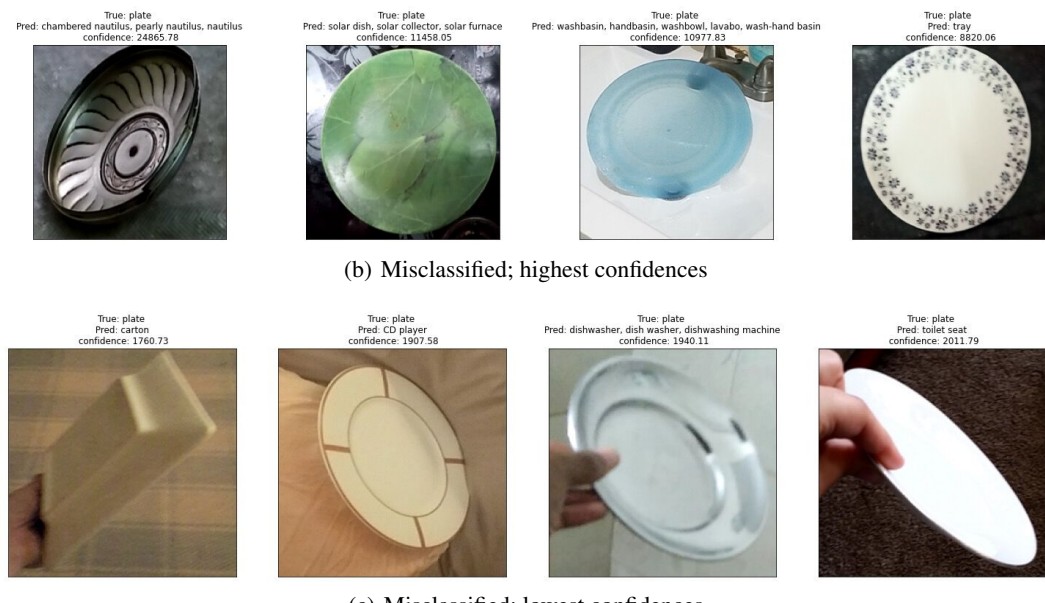

(b) Misclassified; highest confidences

(c) Misclassified; lowest confidences

**Figure 17:** Correctly classified and misclassified examples from the Plate class by the ResNet model.

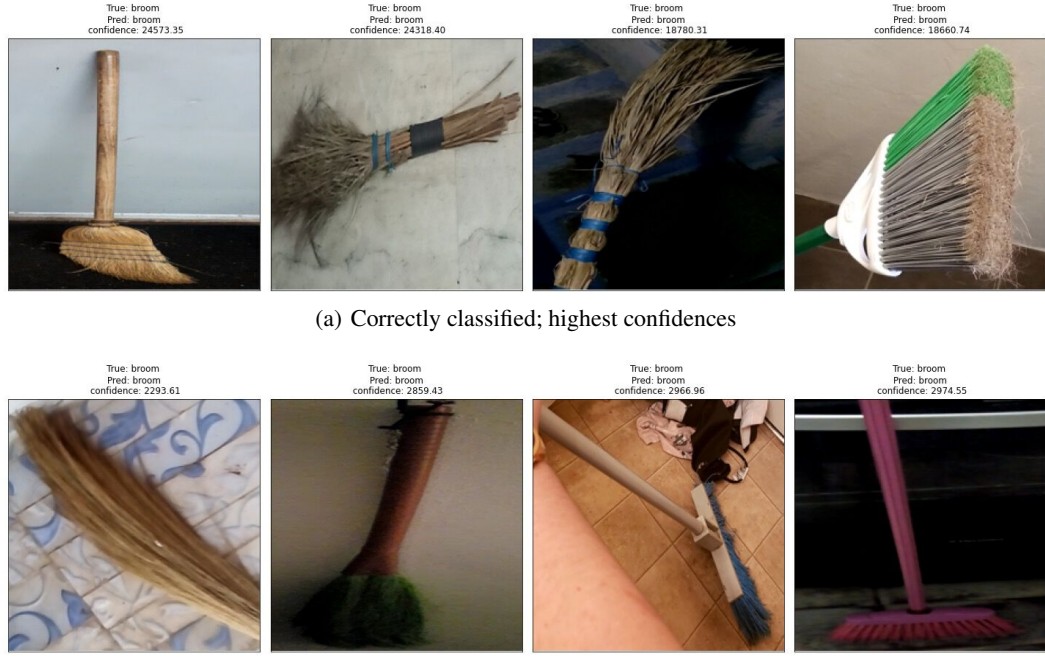

(a) Correctly classified; highest confidences

(b) Correctly classified; lowest confidences

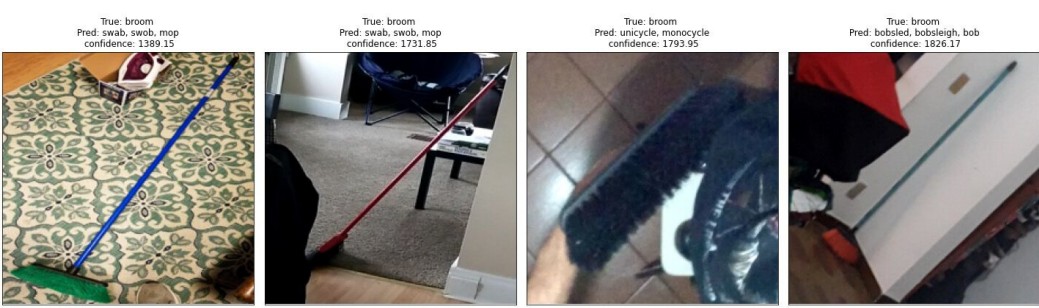

(c) Misclassified; highest confidences

(d) Misclassified; lowest confidences

Figure 18: Correctly classified and misclassified examples from the Broom class by the ResNet model.

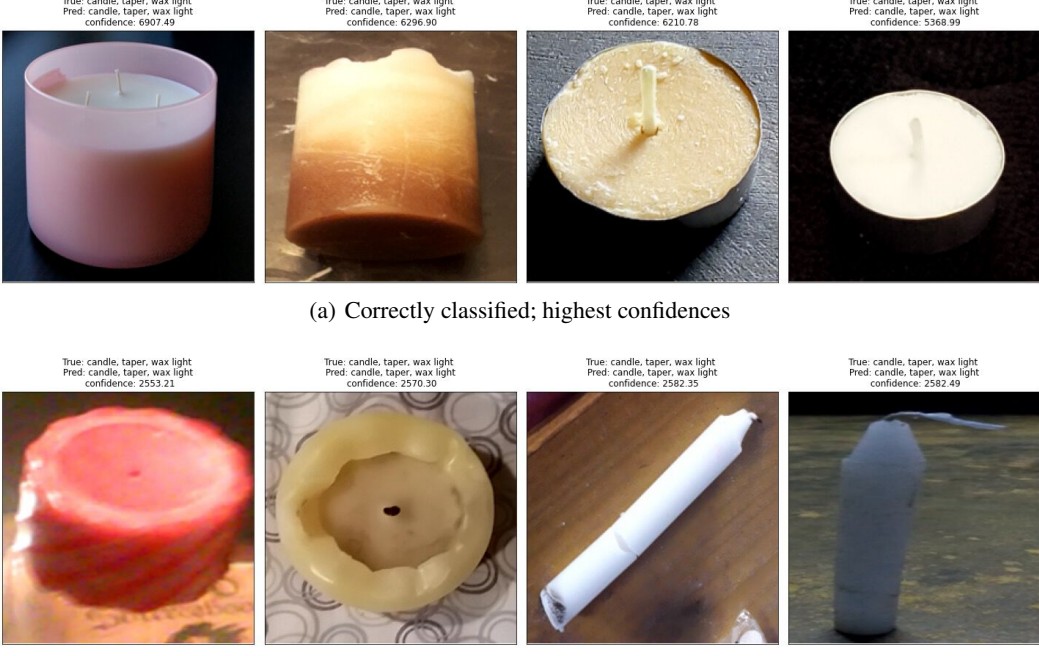

(a) Correctly classified; highest confidences

(b) Correctly classified; lowest confidences

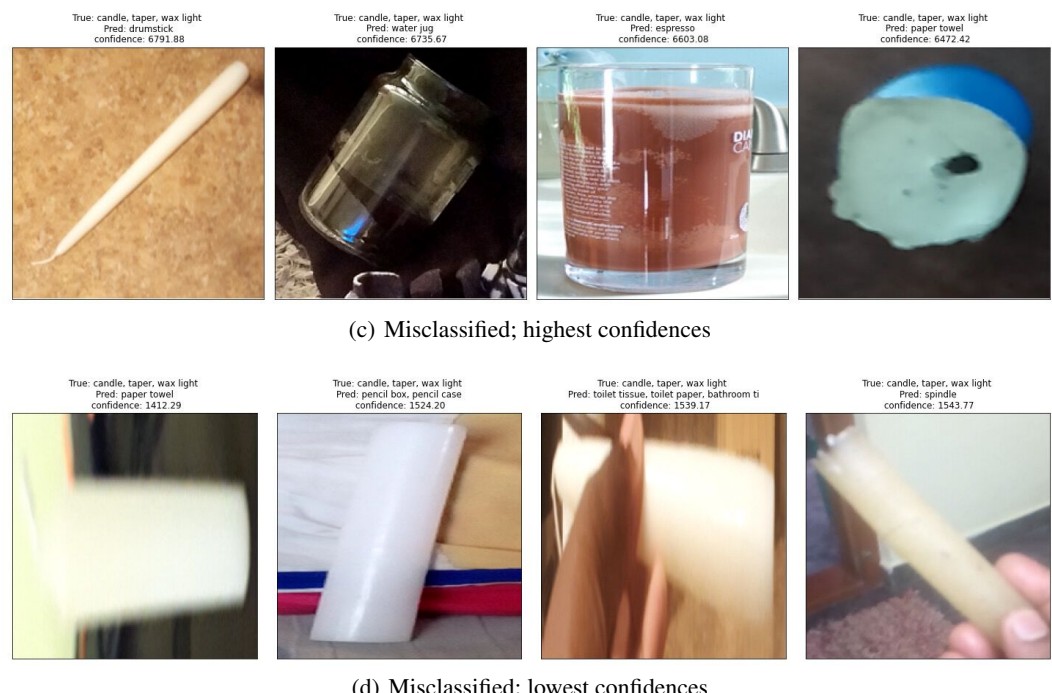

(c) Misclassified; highest confidences

(d) Misclassified; lowest confidences

**Figure 19:** Correctly classified and misclassified examples from the Candle class by the ResNet model.

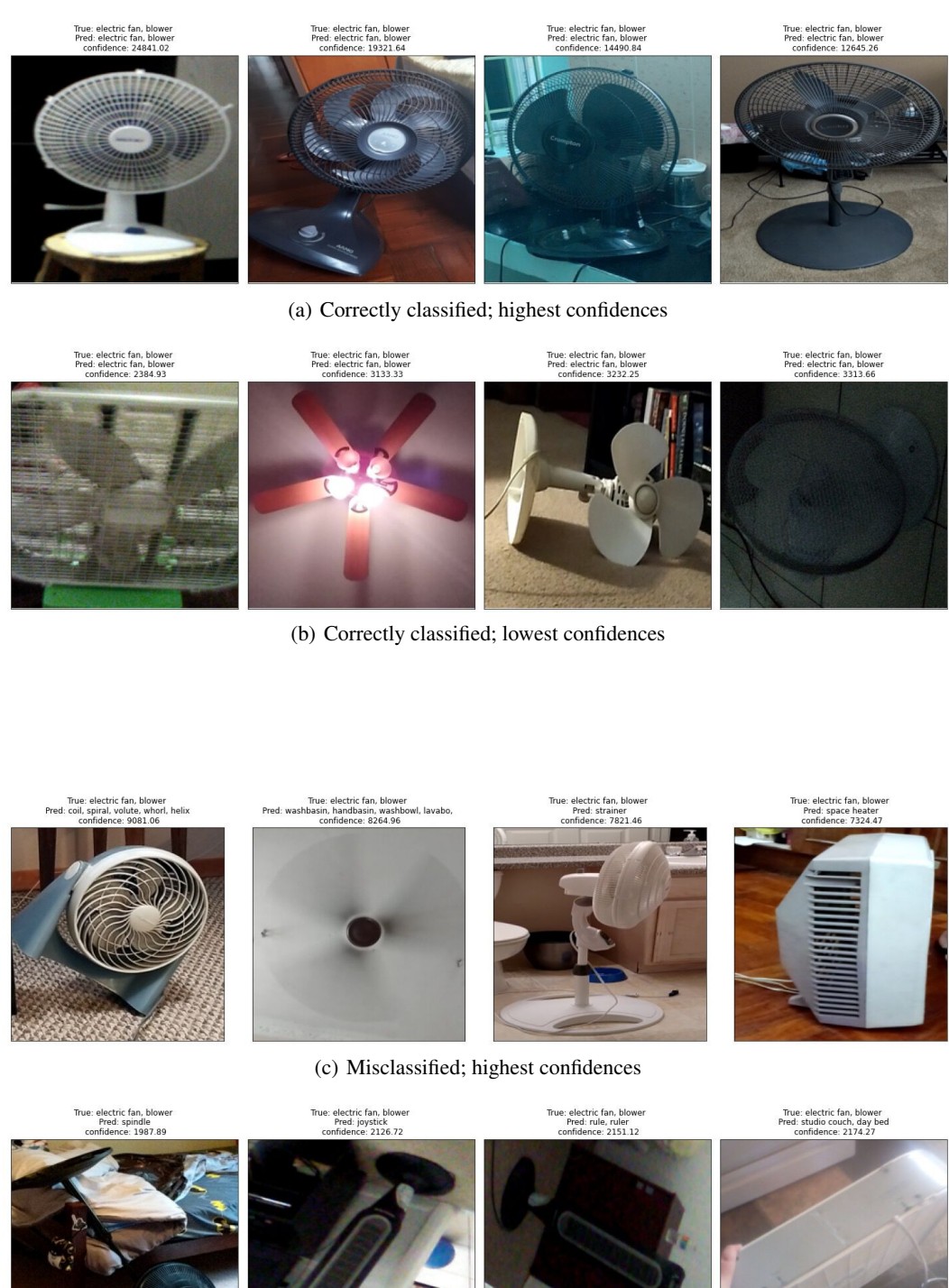

Figure 20: Correctly classified and misclassified examples from the Fan class by the ResNet model.

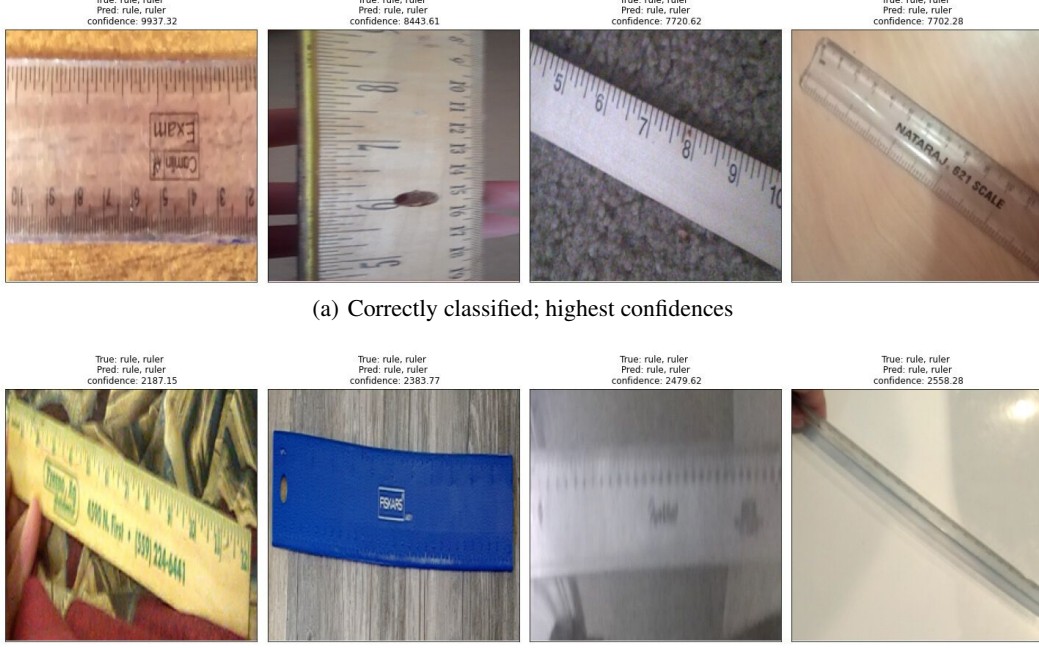

(a) Correctly classified; highest confidences

(b) Correctly classified; lowest confidences

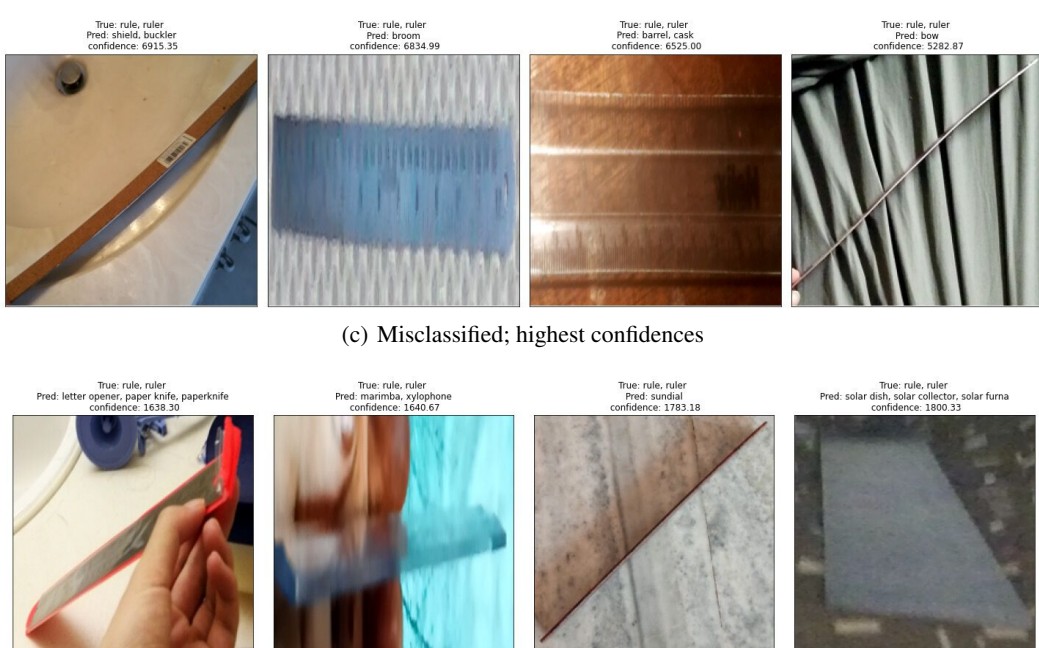

(c) Misclassified; highest confidences

(d) Misclassified; lowest confidences

**Figure 21:** Correctly classified and misclassified examples from the Ruler class by the ResNet model.

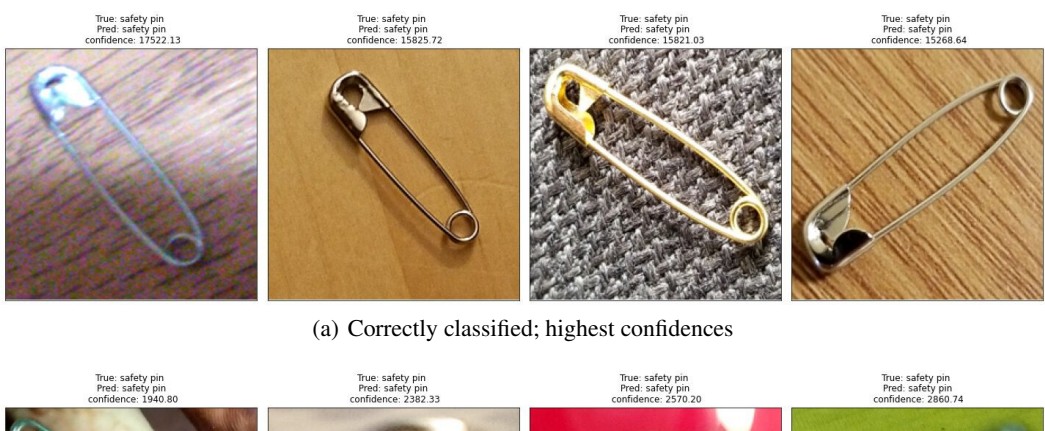

(a) Correctly classified; highest confidences

(b) Correctly classified; lowest confidences

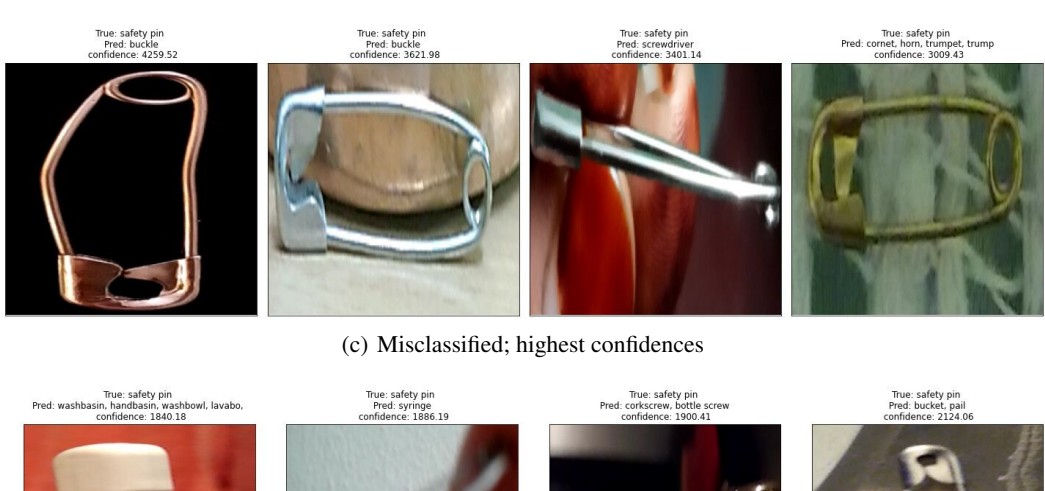

(c) Misclassified; highest confidences

(d) Misclassified; lowest confidences

**Figure 22:** Correctly classified and misclassified examples from the Safety-pin class by the ResNet model.

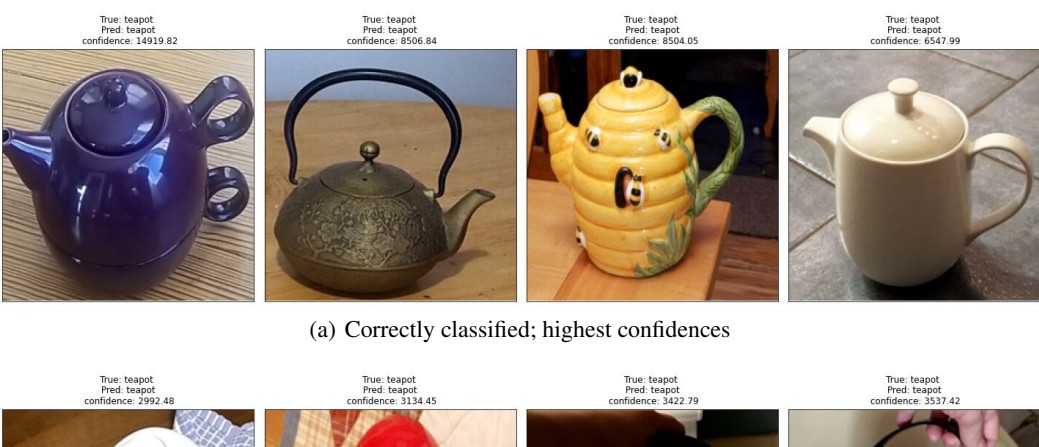

(a) Correctly classified; highest confidences

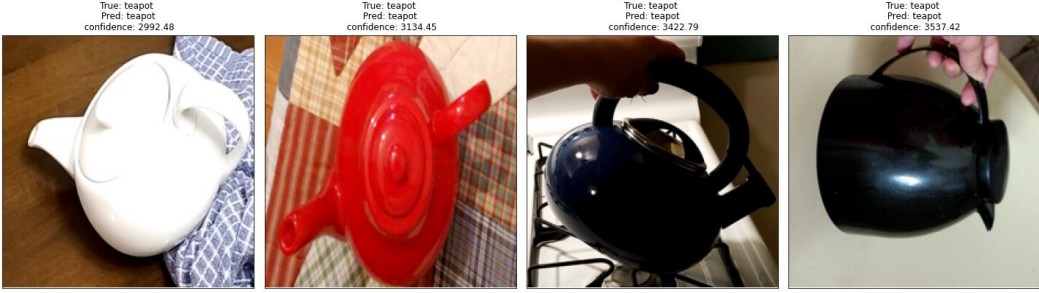

(b) Correctly classified; lowest confidences

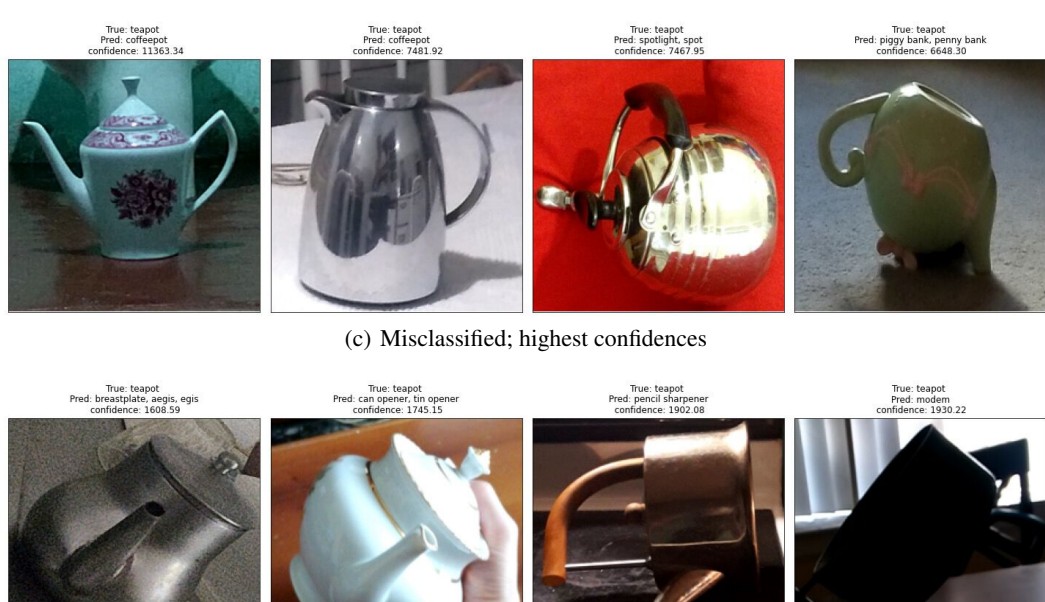

(c) Misclassified; highest confidences

(d) Misclassified; lowest confidences

**Figure 23:** Correctly classified and misclassified examples from the Teapot class by the ResNet model.

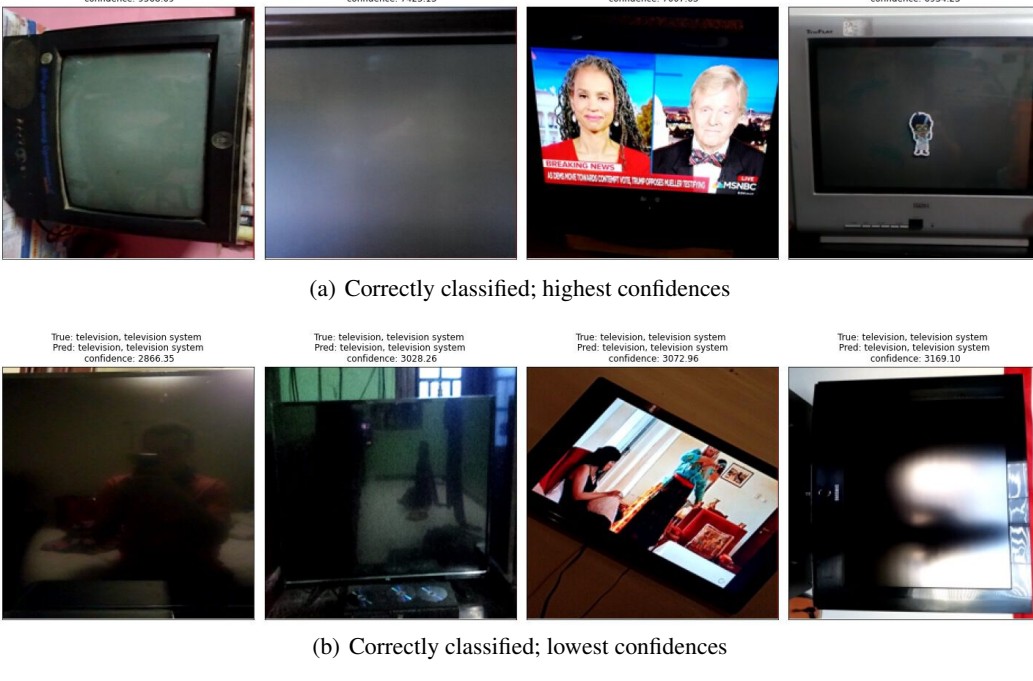

(a) Correctly classified; highest confidences

(b) Correctly classified; lowest confidences

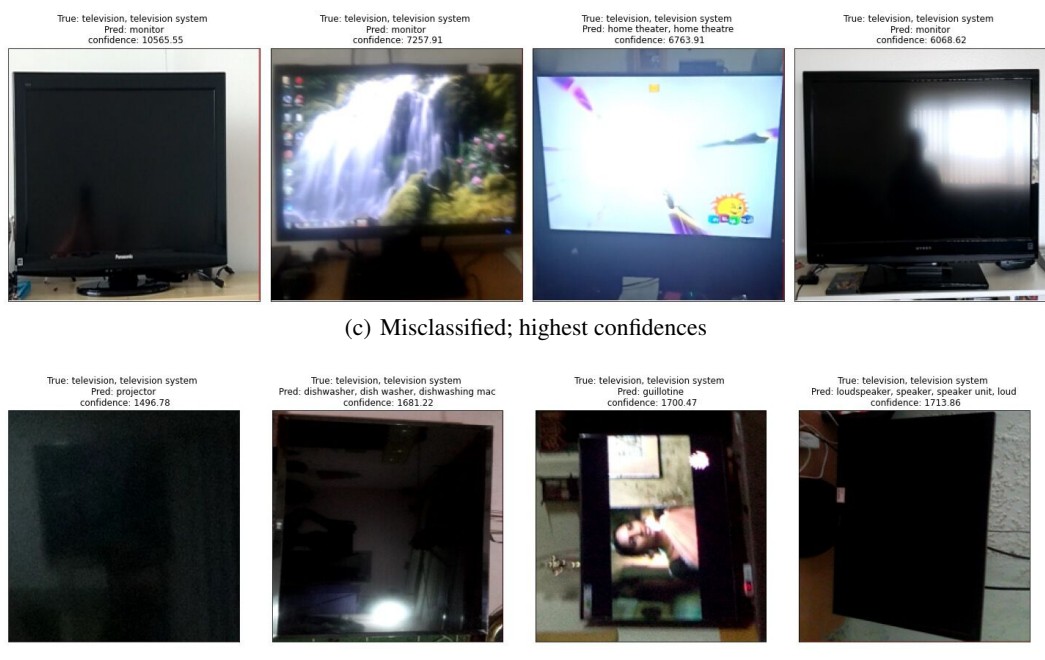

(c) Misclassified; highest confidences

(d) Misclassified; lowest confidences

**Figure 24:** Correctly classified and misclassified examples from the TV class by the ResNet model.

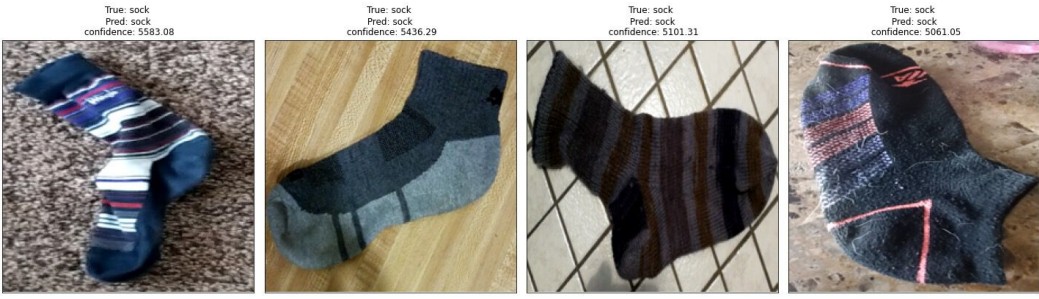

(a) Correctly classified; highest confidences

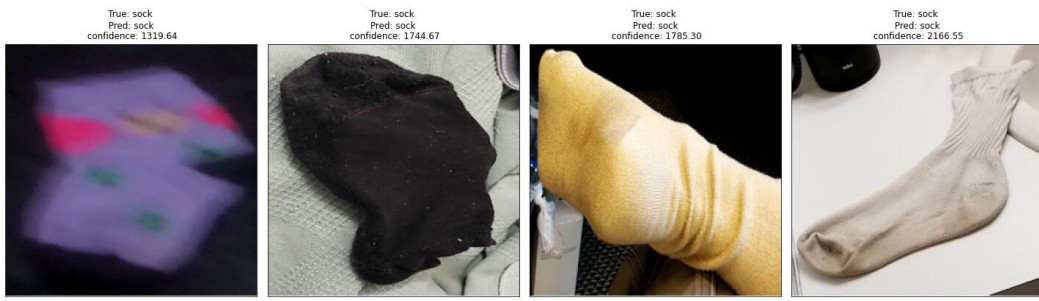

(b) Correctly classified; lowest confidences

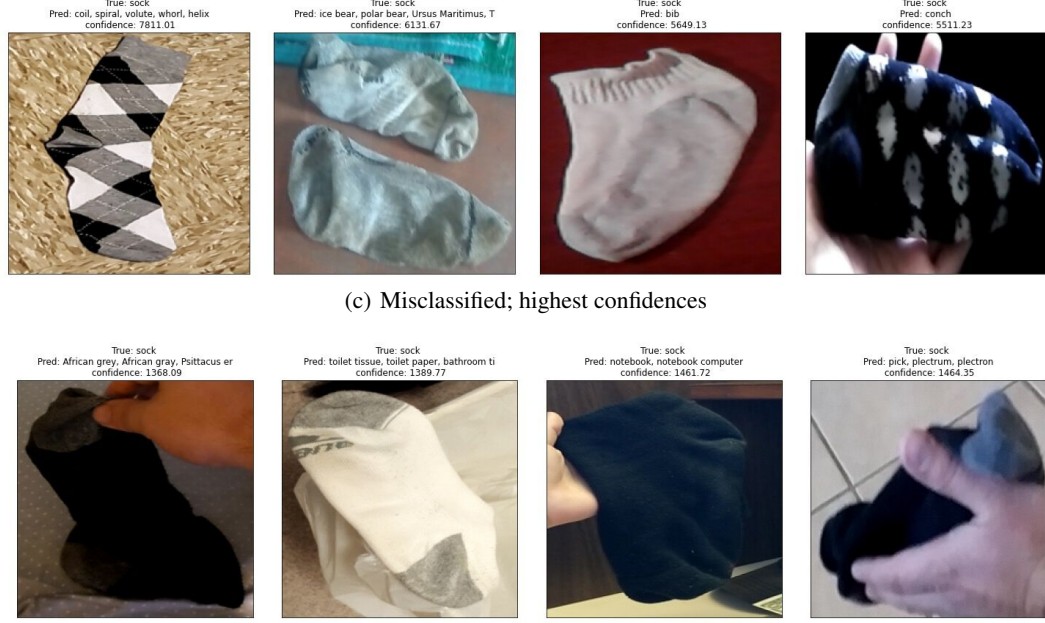

(c) Misclassified; highest confidences

(d) Misclassified; lowest confidences

**Figure 25:** Correctly classified and misclassified examples from the Sock class by the ResNet model.

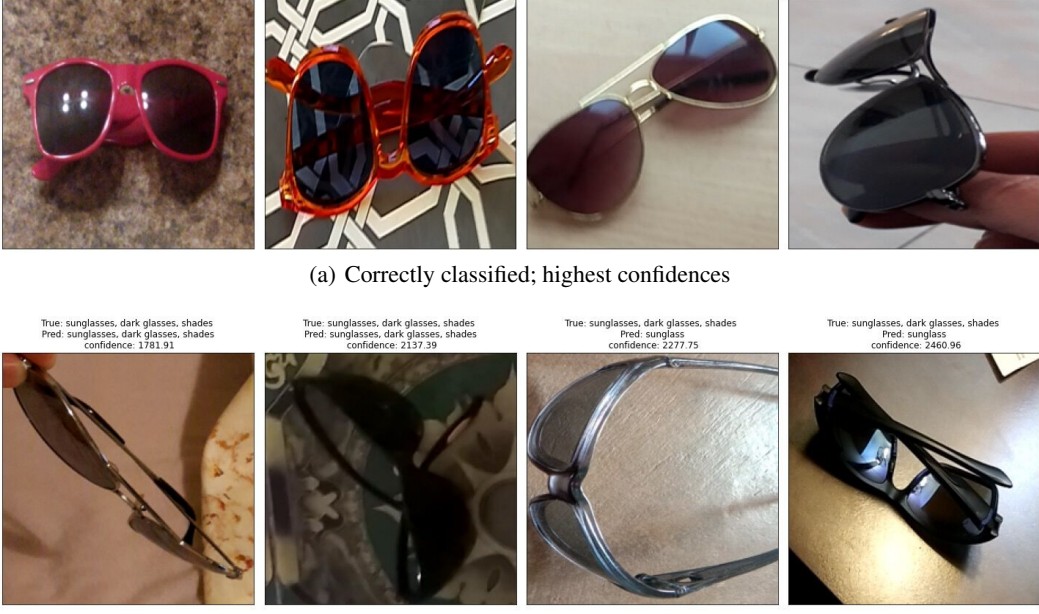

(a) Correctly classified; highest confidences

(b) Correctly classified; lowest confidences

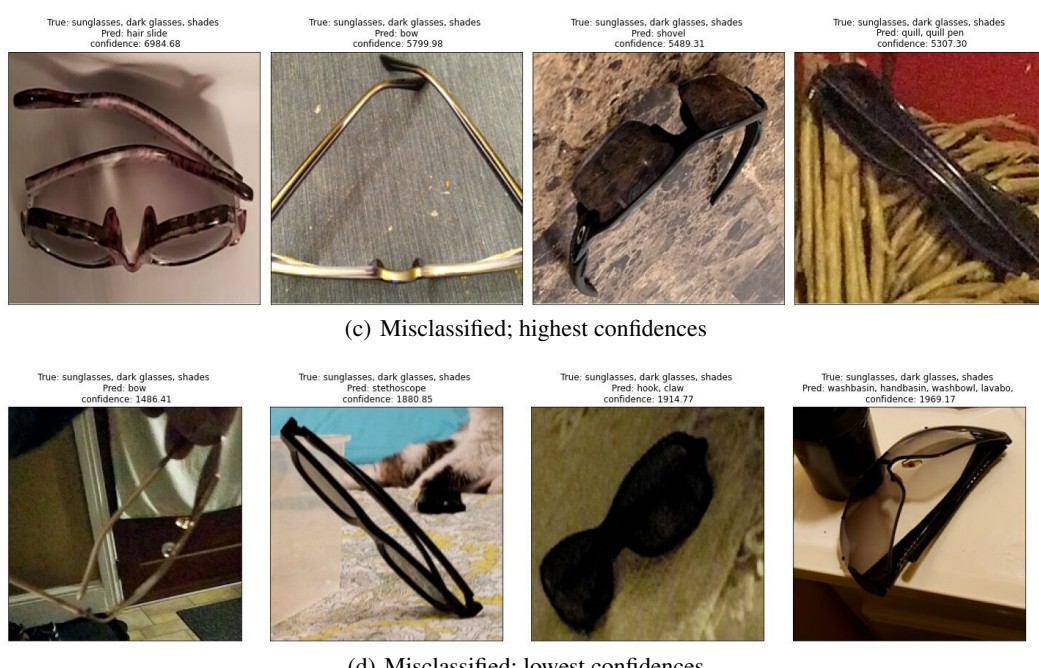

(c) Misclassified; highest confidences

(d) Misclassified; lowest confidences

**Figure 26:** Correctly classified and misclassified examples from the Sunglasses class by the ResNet model.

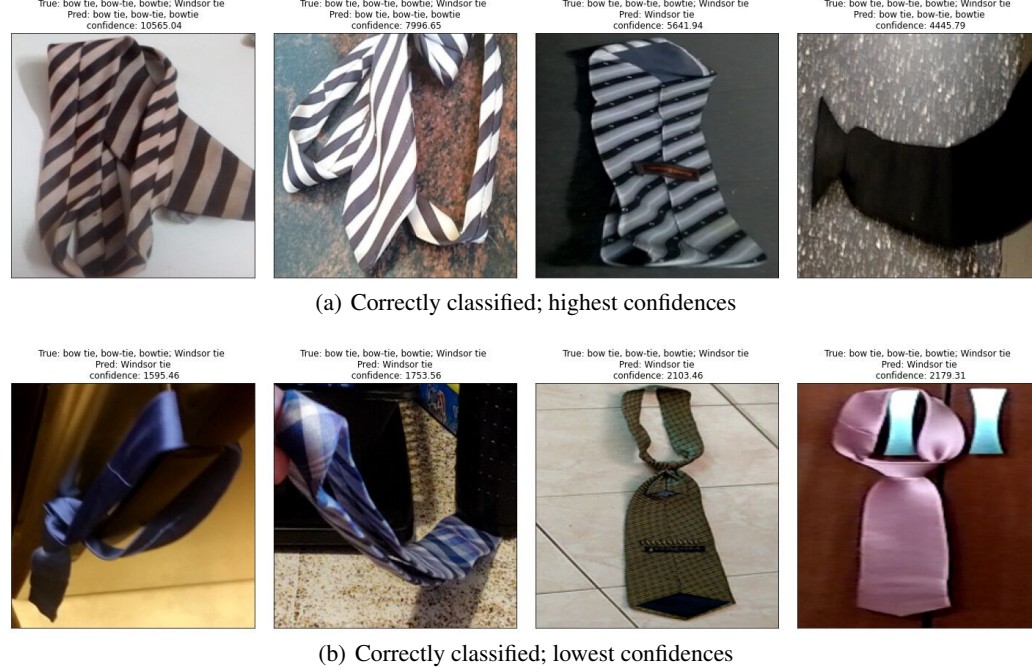

(a) Correctly classified; highest confidences

(b) Correctly classified; lowest confidences

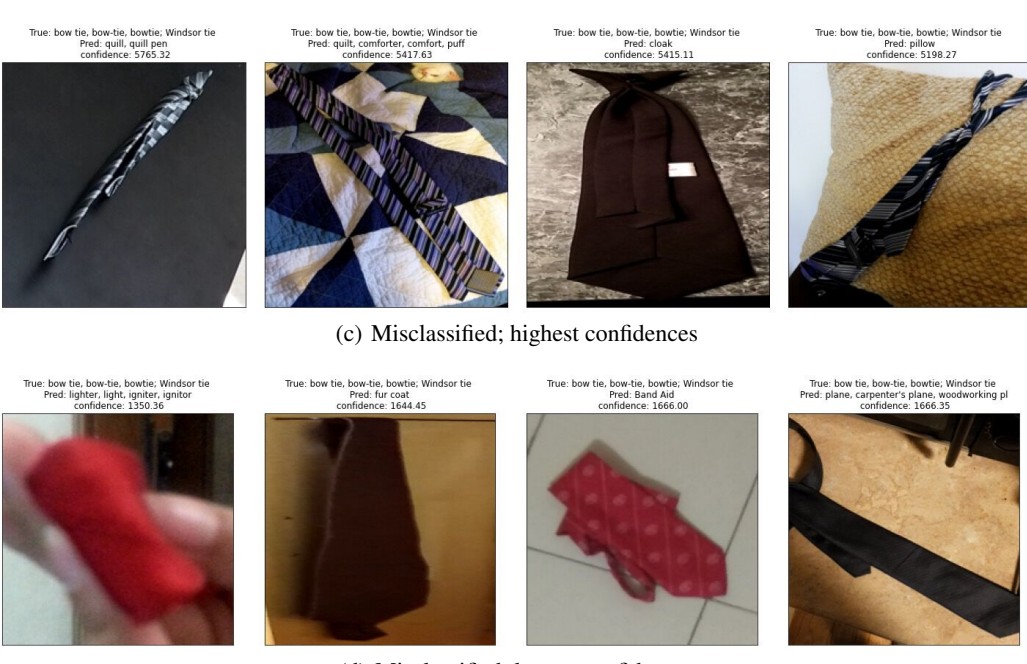

(c) Misclassified; highest confidences

(d) Misclassified; lowest confidences

**Figure 27:** Correctly classified and misclassified examples from the Tie class by the ResNet model.

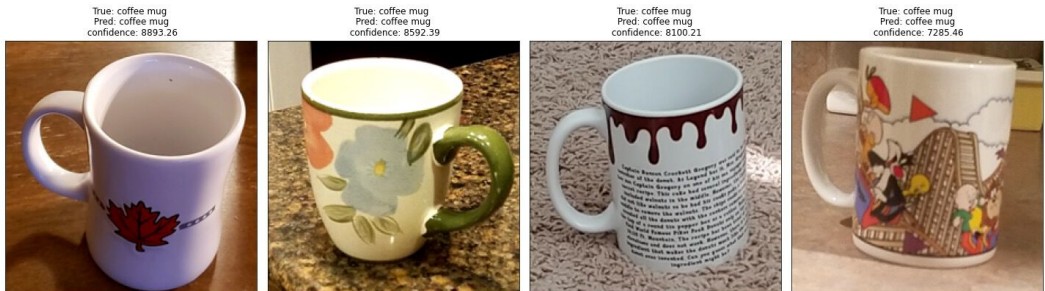

(a) Correctly classified; highest confidences

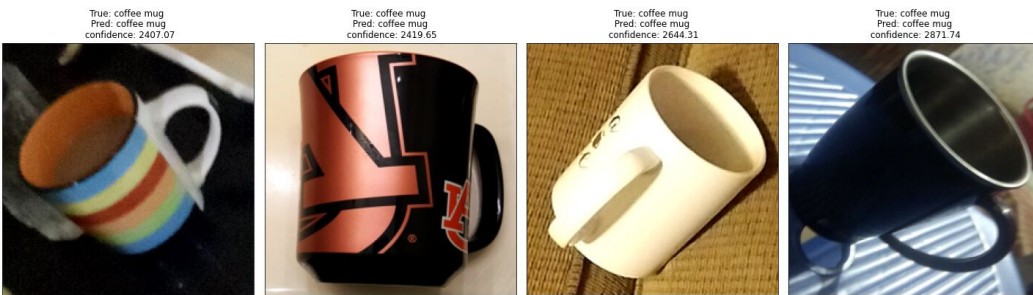

(b) Correctly classified; lowest confidences

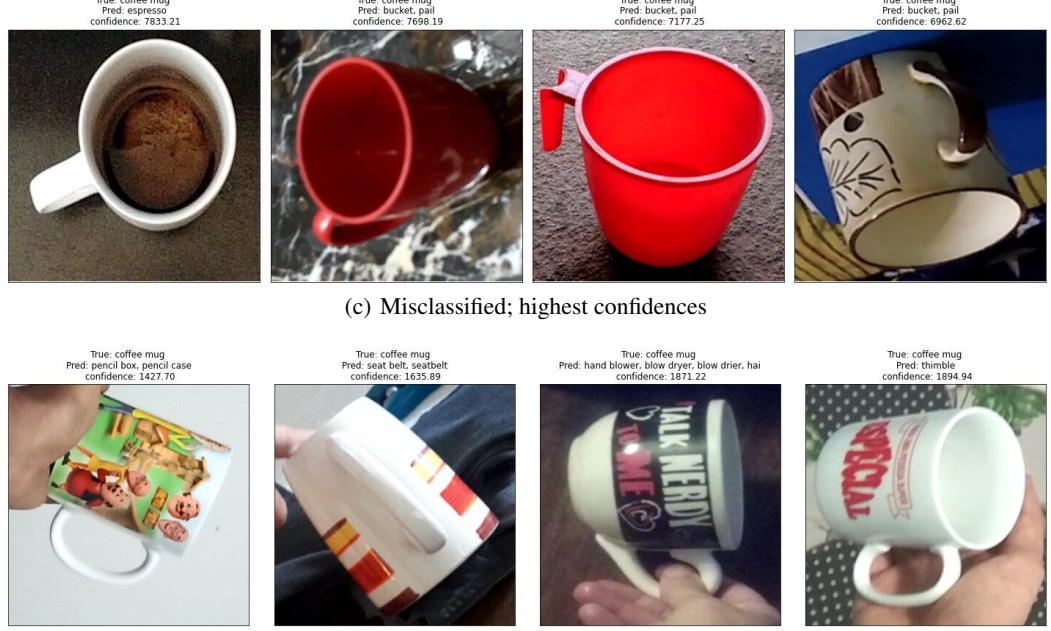

(c) Misclassified; highest confidences

(d) Misclassified; lowest confidences

**Figure 28:** Correctly classified and misclassified examples from the Mug class by the ResNet model.

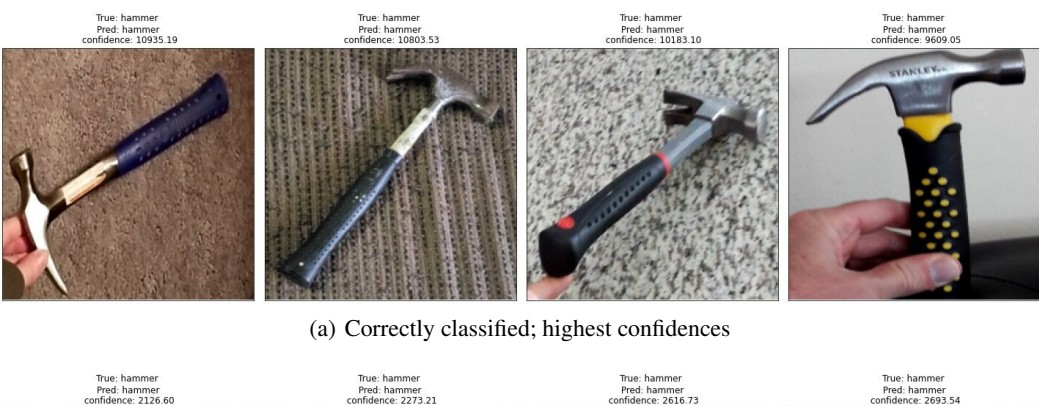

(a) Correctly classified; highest confidences

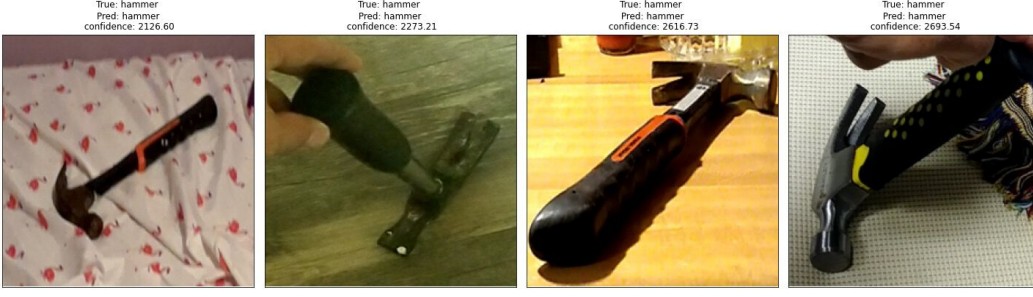

(b) Correctly classified; lowest confidences

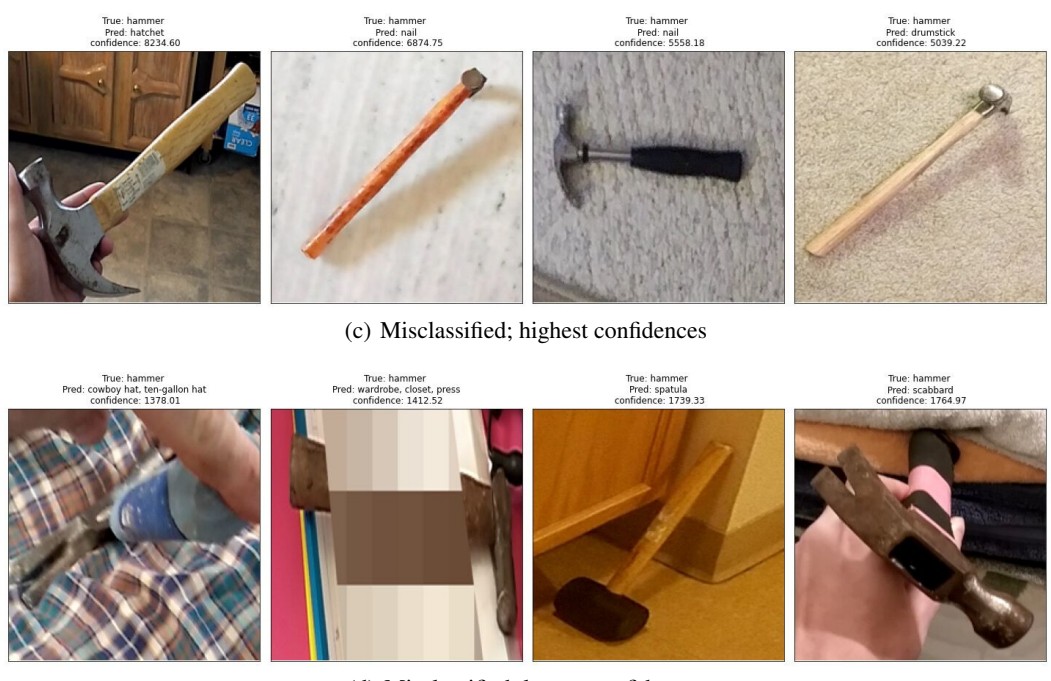

(c) Misclassified; highest confidences

(d) Misclassified; lowest confidences

**Figure 29:** Correctly classified and misclassified examples from the Hammer class by the ResNet model.

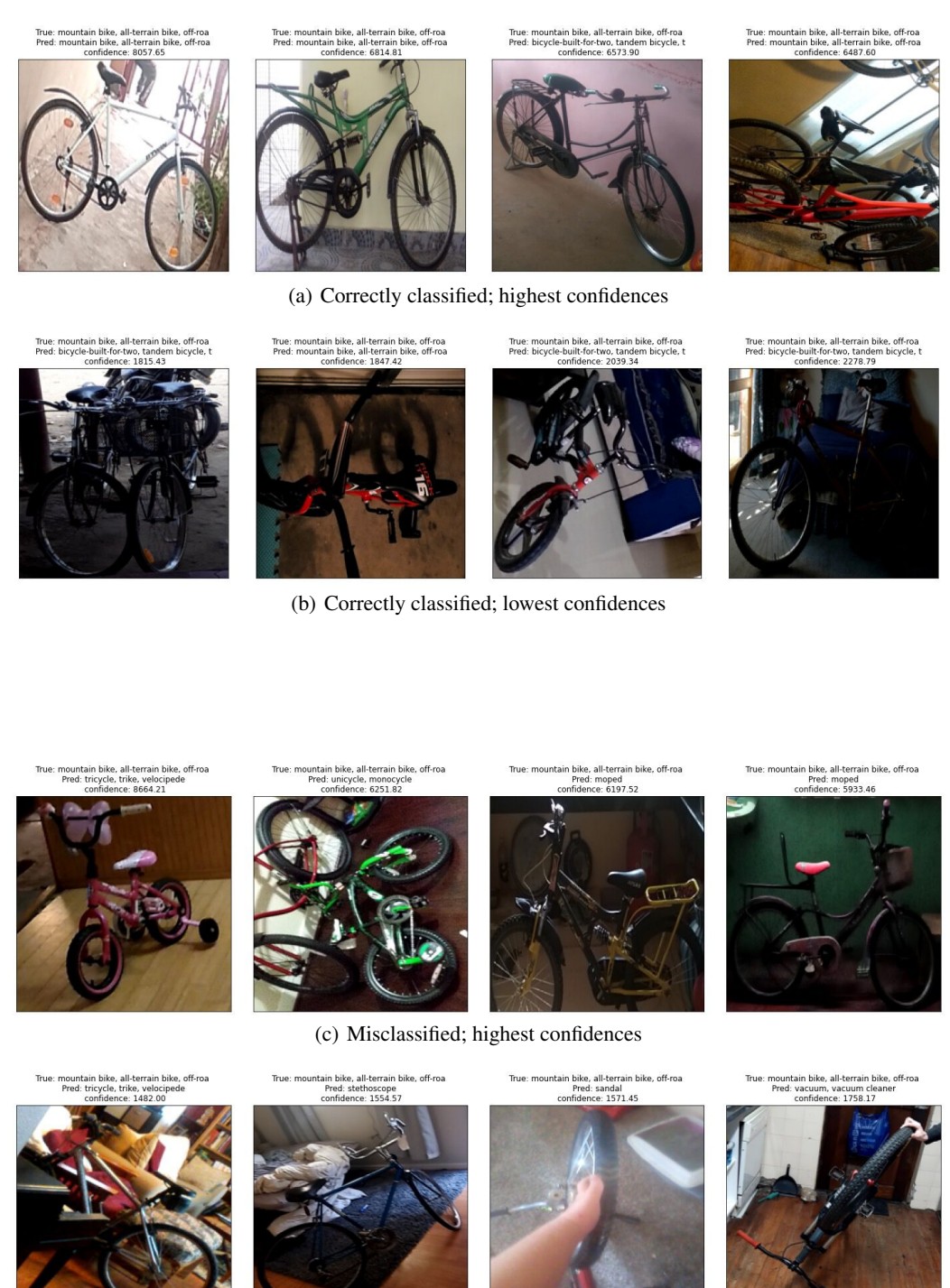

Figure 30: Correctly classified and misclassified examples from the Bicycle class by the ResNet model.

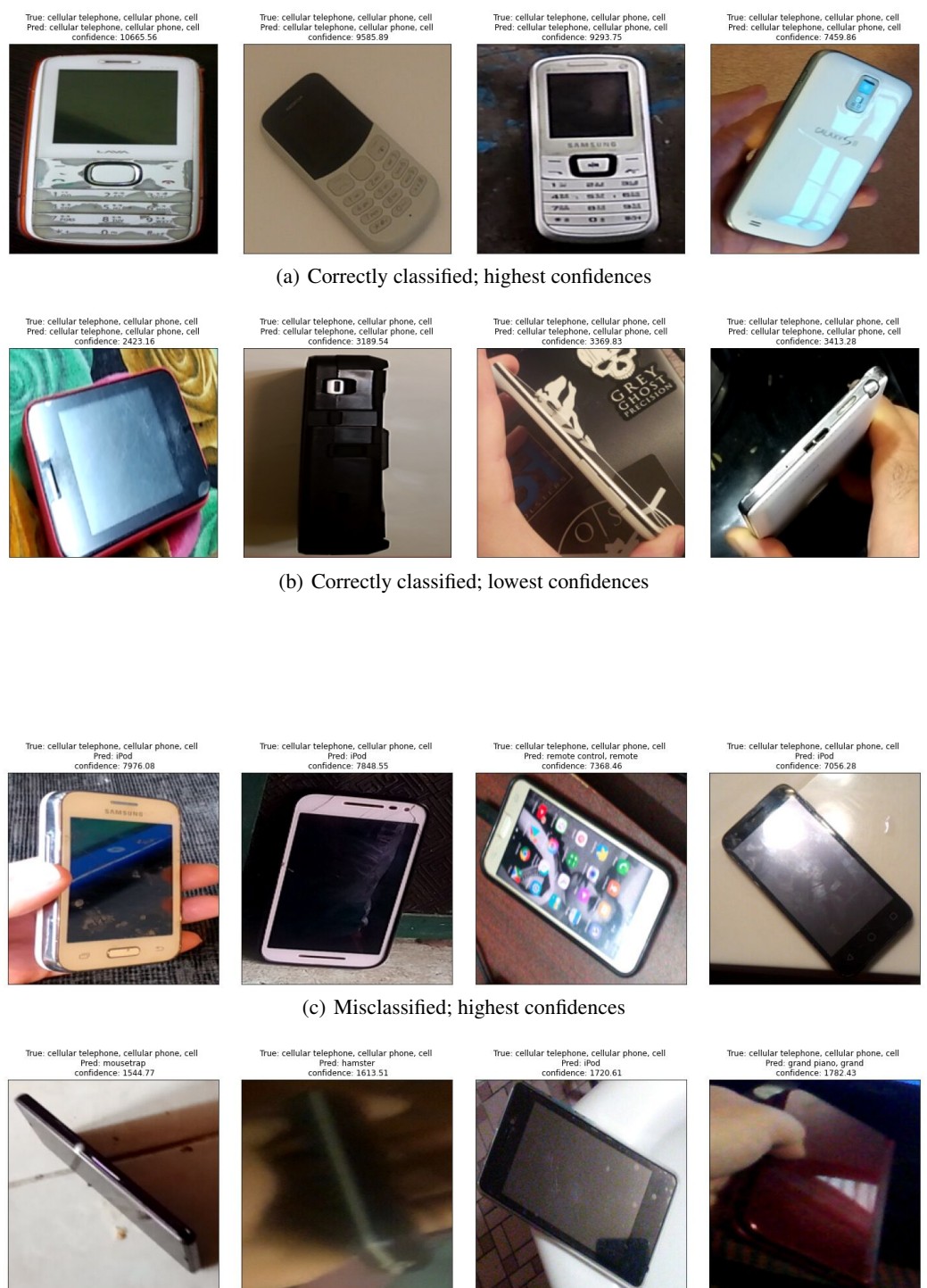

Figure 31: Correctly classified and misclassified examples from the Cellphone class by the ResNet model.

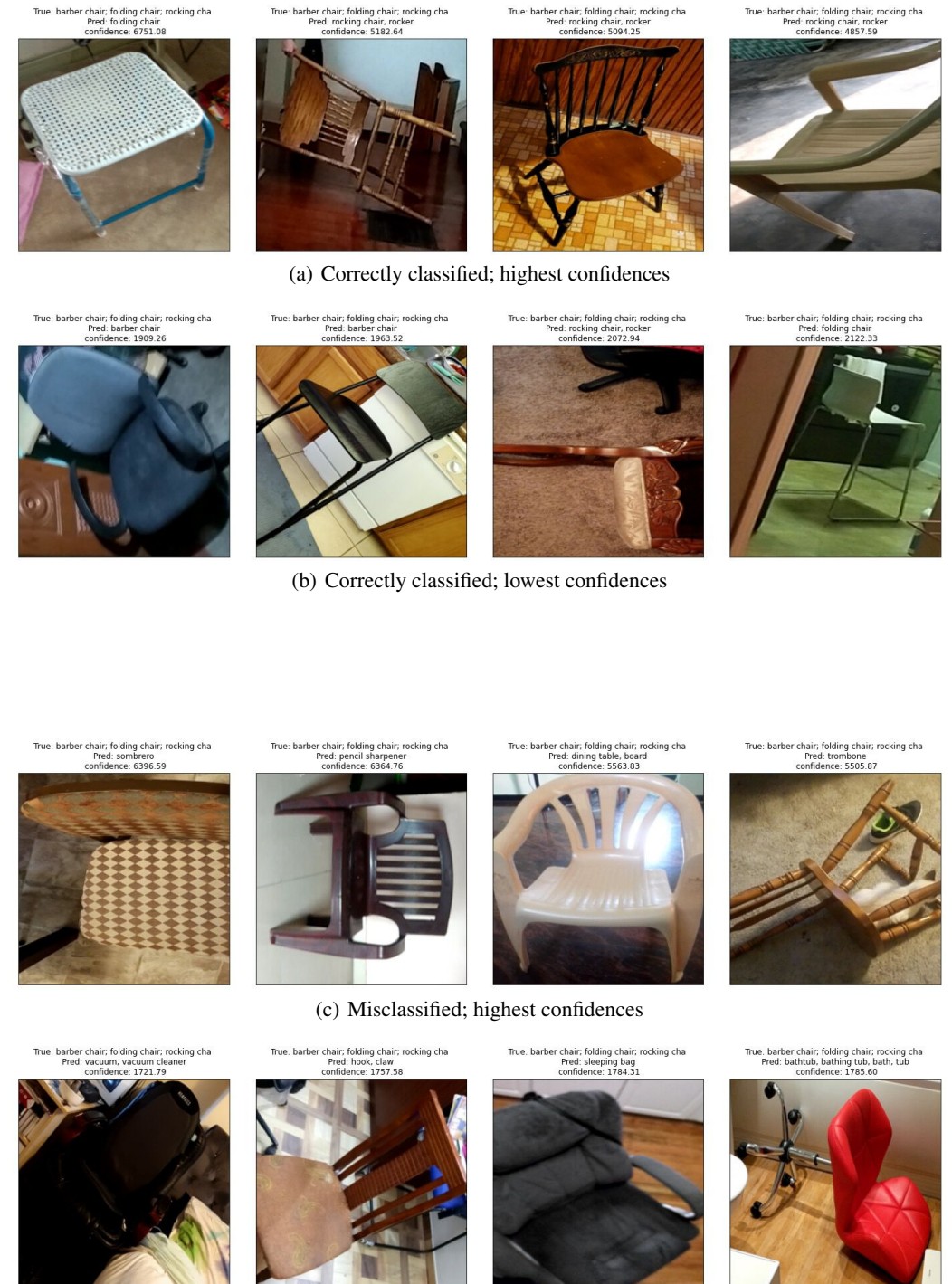

Figure 32: Correctly classified and misclassified examples from the Chair class by the ResNet model.

# D    SOME CHALLENGING EXAMPLES FOR HUMANS

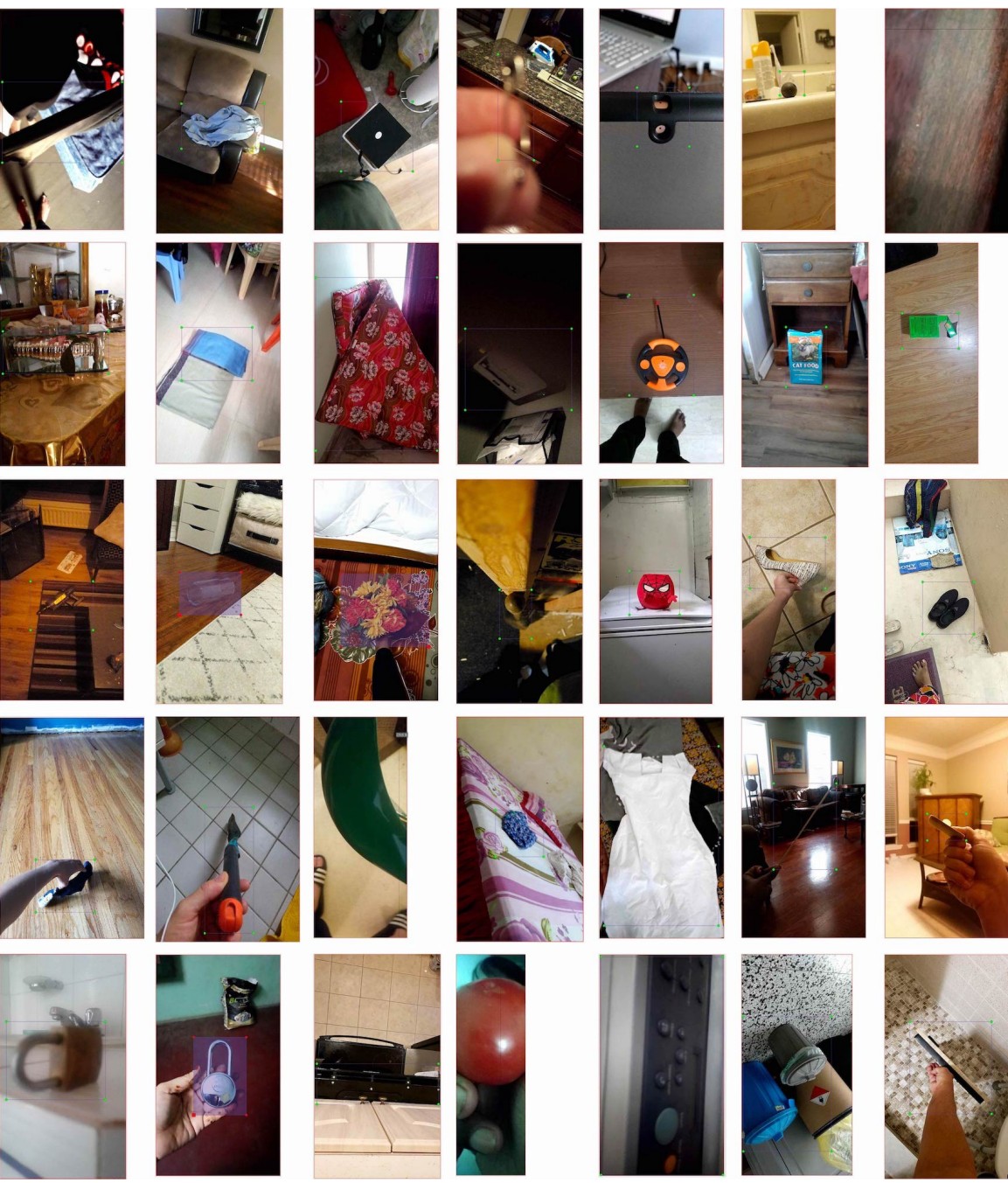

**Figure 33:** A selection of challenging objects that are hard to be recognized by humans. Can you guess the category of the annotated objects in these images? Keys are as follows:
row 1: *(skirt, skirt, desk lamp, safety pin, still camera, spatula, tray)*,
row 2: *(vase, pillow, sleeping bag, printer, remote control, pet food container, detergent)*,
row 3: *(vacuum cleaner, vase, vase, shovel, stuffed animal, sandal, sandal)*,
row 4: *(sock, shovel, shovel, skirt, skirt, match, spatula)*,
row 5: *(padlock, padlock, microwave, orange, printer, trash bin, tray)*

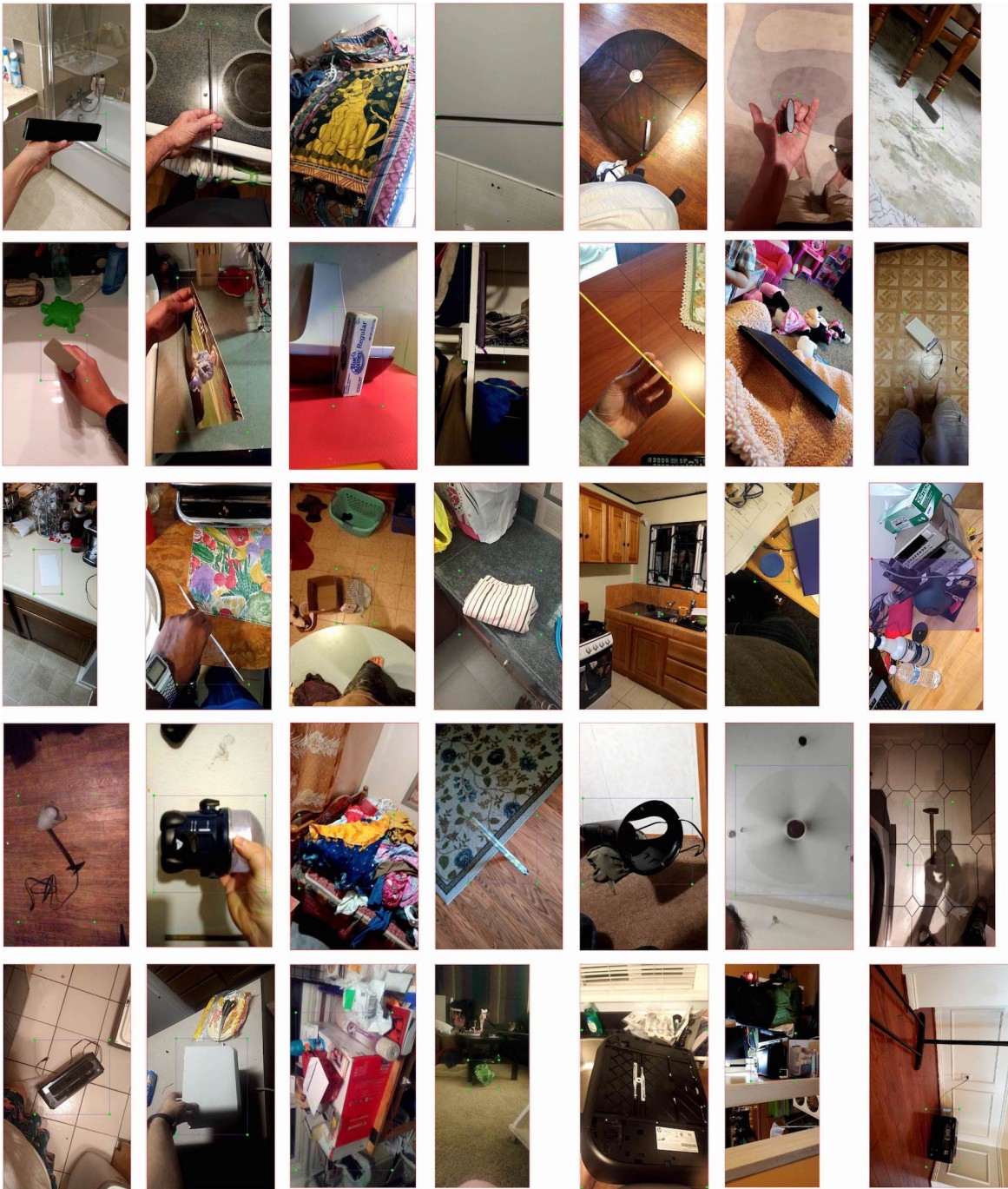

**Figure 34:** A selection of challenging objects that are hard to be recognized by humans (continued from above). Can you guess the category of the annotated objects in these images? Keys are as follows:

row 1: *(remote control, ruler, full sized towel, ruler, remote control, remote control, remote control)*,

row 2: *(remote control, calendar, butter, bookend, ruler, tray, desk lamp)*,

row 3: *(envelope, envelope, drying rack for dishes, full sized towel, drying rack for dishes, drinking cup, desk lamp)*,

row 4: *(desk lamp, desk lamp, dress, tennis racket, fan, fan, hammer)*,

row 5: *(printer, toaster, printer, helmet, printer, printer, printer)*

# E ANNOTATION ISSUES IN OBJECT RECOGNITION DATASETS

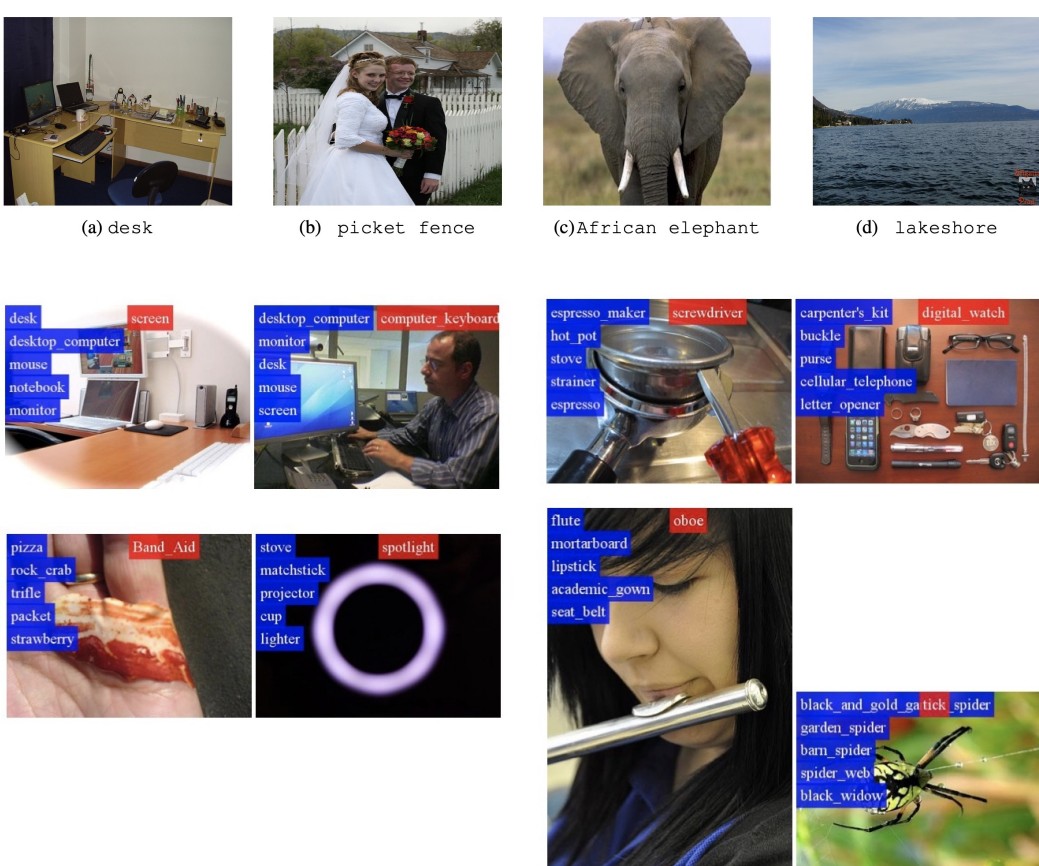

**Figure 35: Top:** Some examples from the ImageNet dataset highlighting why multi-label annotations are necessary: (a) Multiple objects. Here, the image label is desk but screen, monitor, coffee mug and many more objects in the scene could count as correct labels, (b) Non-salient object. A scene where the target label picket fence is counter intuitive because it appears in the image background, while classes groom, bow-tie, suit, gown, and possibly hoopskirt are more prominently displayed in the foreground. c) Synonym or subset relationships: This image has ImageNet label African elephant, but can be labeled tusker as well, because every African elephant with tusks is a tusker. d) Unclear images: This image is labeled lake shore, but could also be labeled seashore as there is not enough information in the scene to distinguish the water body between a lake or sea. Image compiled from (Shankar et al., 2020). **Bottom:** Some annotation problems in the ImageNet dataset. Example images along with their ground truth (GT) labels (red) and predicted classes (PC) by a model (blue) are shown. Top-left) Similar labels, Top-right) non-salient GT, Bottom-left) challenging images, and Bottom-right) incorrect GT. Please see Lee et al. (2017) for details.

# F  ANALYSING MODEL ROBUSTNESS OVER NATURALLY DISTORTED IMAGES

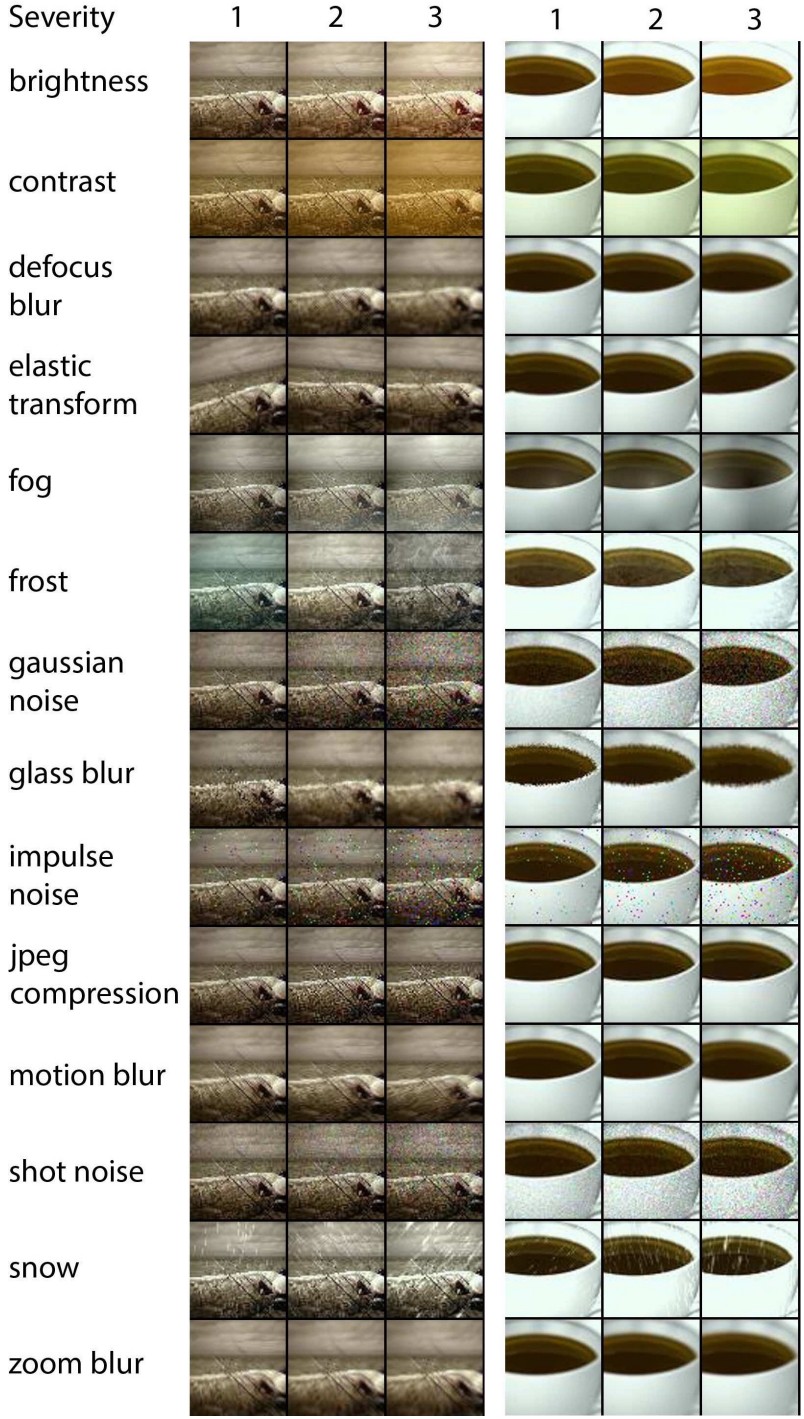

**Figure 36:** Sample images alongside their corruptions at 3 severity levels.

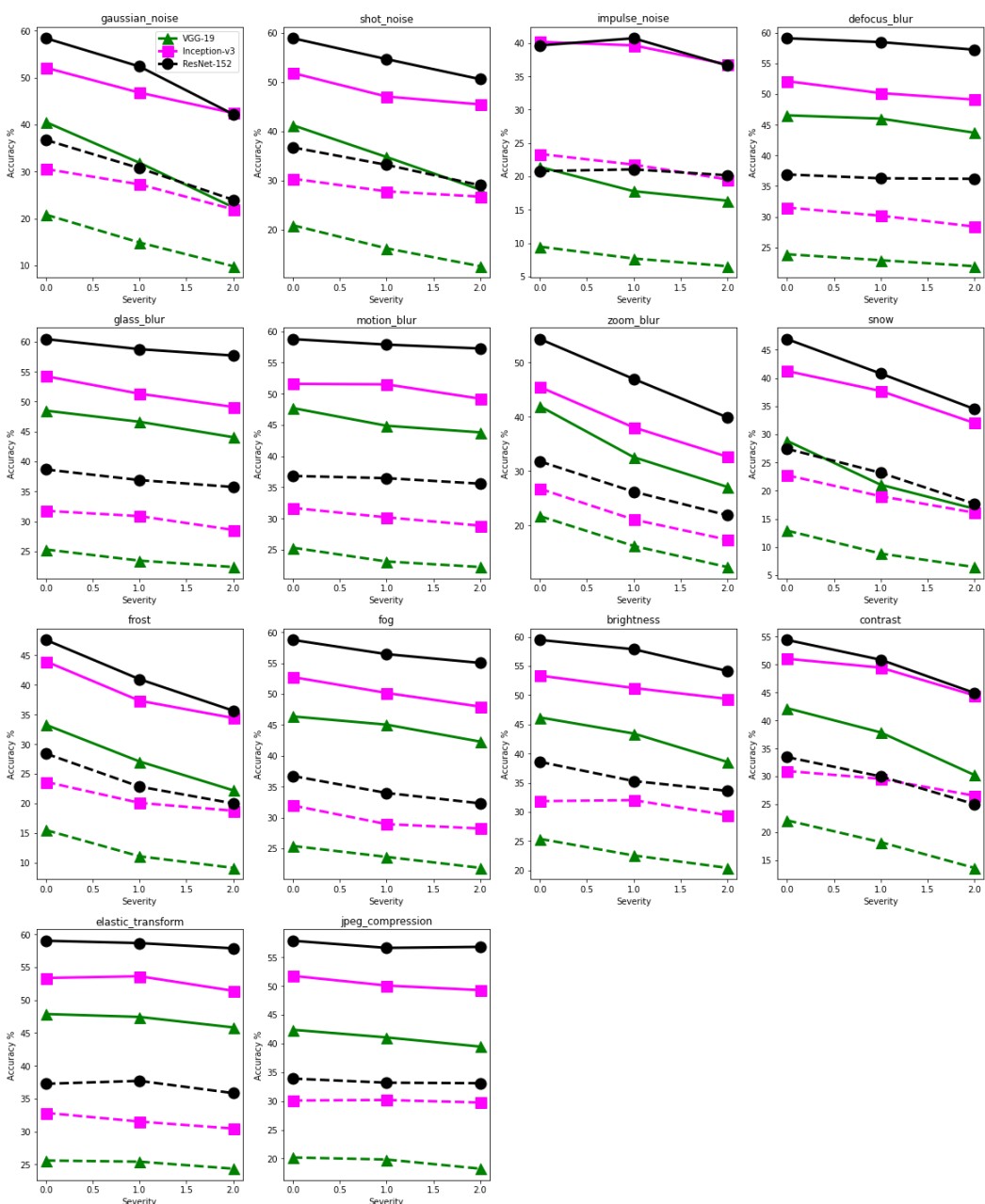

**Figure 37:** Top-1 (dashed lines) and Top-5 (solid lines) accuracy of models over 14 natural image distortions at 3 severity levels (using object bounding box).

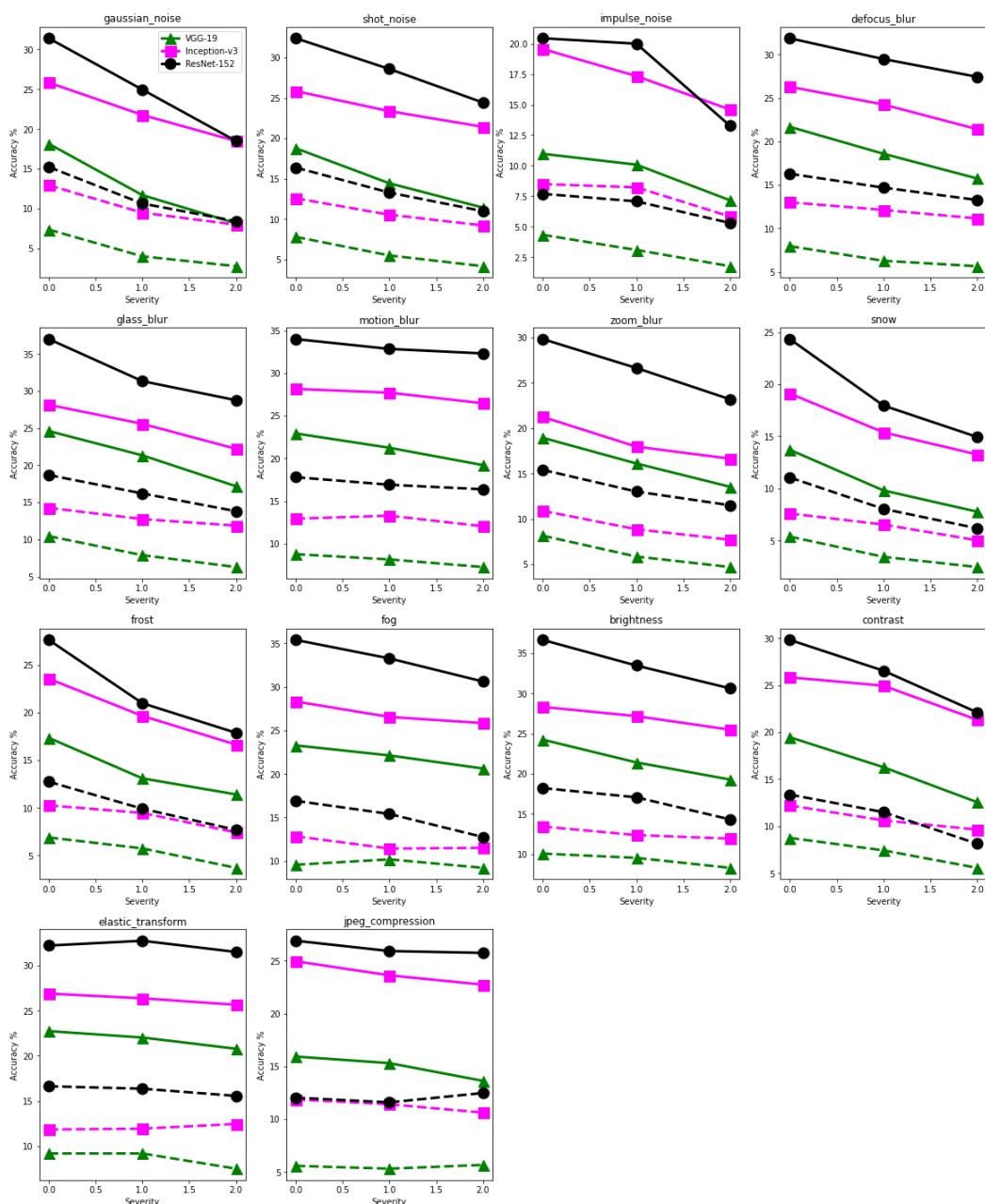

**Figure 38:** Top-1 (dashed lines) and Top-5 (solid lines) accuracy of models over 14 natural image distortions at 3 severity levels (using full image).

# G  ADVERSARIAL DEFENSE USING FOREGROUND DETECTION ON MNIST AND FASHION MNIST

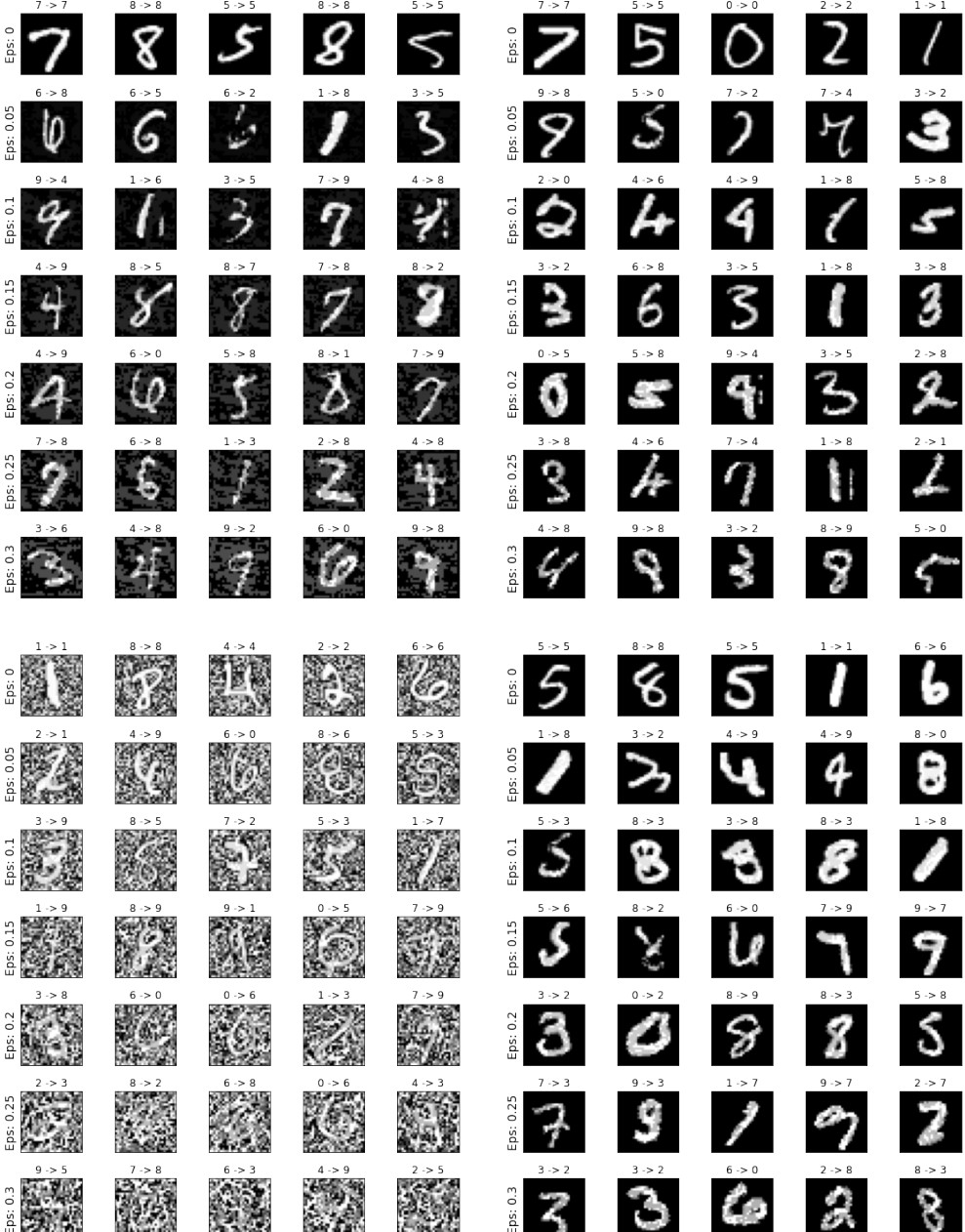

**Figure 39:** Top-left) Model, trained on clean MNIST data, is attacked by FGSM, Top-right) Foreground detection over a model that has been trained on clean data. Bottom-left) Model, trained on noisy MNIST data, is attacked by FGSM, Bottom-right) Foreground detection of a model that has been trained on noisy data. Noisy data is created by overlaying an object over white noise i.e., noise × (1- mask) + object. We find that background subtraction together with edge detection improves robustness.

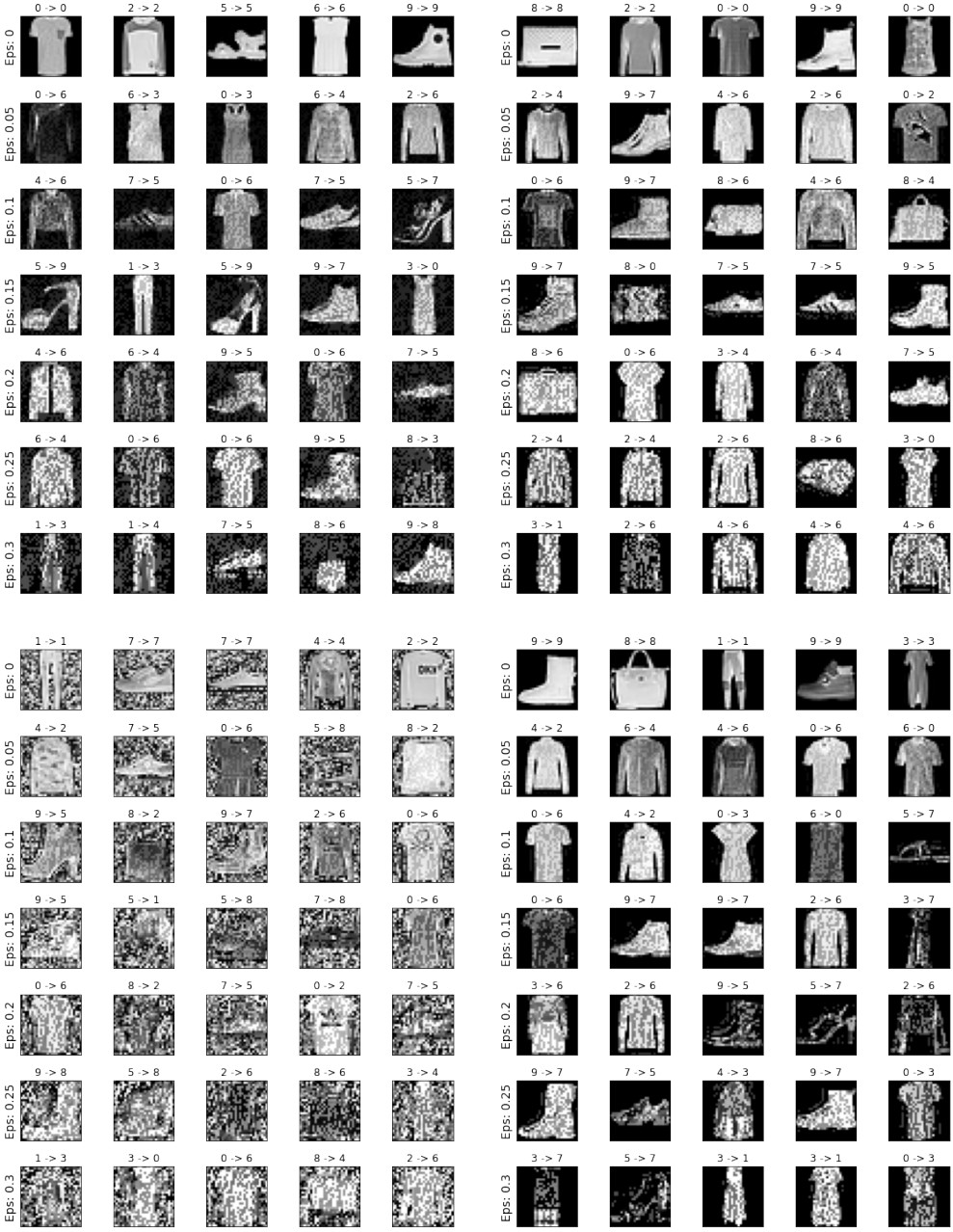

**Figure 40:** Top-left) Model, trained on clean FashionMNIST data, is attacked by FGSM, Top-right) Foreground detection over a model that has been trained on clean data. Bottom-left) Model, trained on noisy FashionMNIST data, is attacked by FGSM, Bottom-right) Foreground detection of a model that has been trained on noisy data. Noisy data is created by overlaying an object over white noise i.e., noise × (1- mask) + object. We find that background subtraction together with edge detection improves robustness.

# H    STATISTICS OF OBJECTNET DATASET

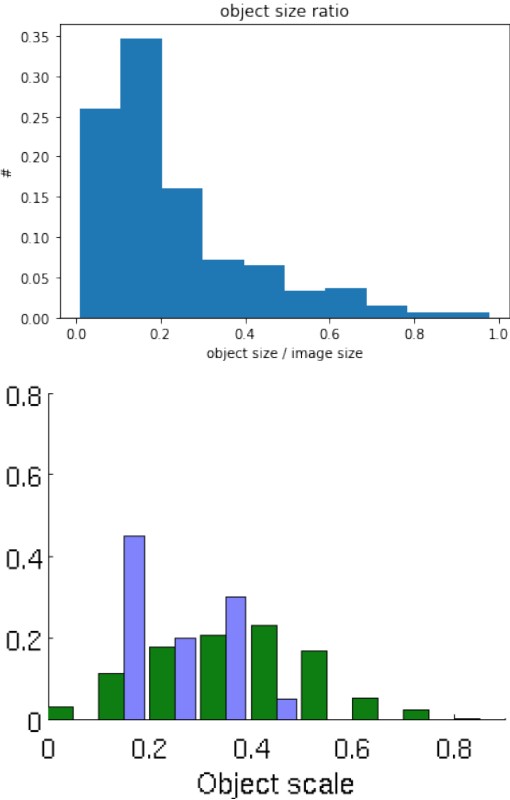

**Figure 41:** Top: Distribution of object scale in ObjectNet dataset (113 annotated classes), Bottom: Distribution object scale in ILSVRC2012-2014 single-object localization (dark green) and PASCAL VOC 2012 (light blue) validation sets. Object scale is fraction of image area occupied by an average object instance. Please see Fig. 16 in Russakovsky et al. (2015) .

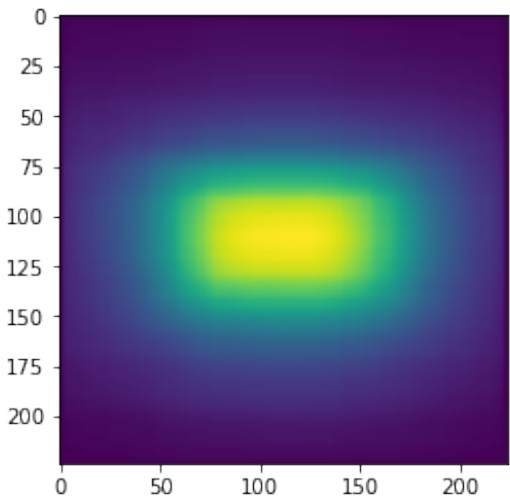

**Figure 42:** Distribution of object location in ObjectNet (113 annotated classes).

