# OpenReview forum: "Contemplating Real-World Object Classification"
_ICLR.cc/2021/Conference — ICLR 2021 Poster_

### Official Review · AnonReviewer2 · 2020-10-27
**Exciting work. Some claim might be flawed, and experiment setting could violate dataset license.**

**Rating:** 6
**Confidence:** 4

**Review:**

This paper revisits ObjectNet dataset closely and found applying classifiers on object bounding box significantly reduces the gap between ImageNet and ObjectNet. The authors further investigates the robustness of CNNs against image perturbations and adversarial attacks, and found limiting the object area to their segmentation mask significantly improves model accuracy (and robustness). Qualitative evaluation is also performed over confident and less-confident / incorrect model predictions and find it correlates with human perception.

Pros:
- More analyses like this paper does would help bridge the gap between ML/CV model performance in staged datasets and real-world scenarios
- The experimentation conducted in this paper is comprehensive, accompanied with many in-depth inspection over a large dataset. The insights drawn from this paper would be invaluable for researchers working in this field.

Cons:
- My major concern with this paper (and the main factor of rating it clear rejection) is the experimental setup used in section 3.3. From authors they "selected images from ten classes of the ObjectNet dataset ... manually annotated the object of interest in each image". Then "Models were trained on 70 images per category".  (also from Figure 39 "In total we annotated 1K images across ten categories of the ObjectNet dataset."). If interpreted correctly, the models are trained on part of ObjectNet images which clearly violates dataset license "ObjectNet may never be used to tune the parameters of any model." (https://objectnet.dev/download.html).

- While appreciating the authors conducting study on model robustness, the conclusion drawn from several experiments seems to confuse "performance" v.s. "robustness", where the former indicates model have better accuracy, and the latter measures how model accuracy varies with increasing noise / perturbations. See more details below.

Some (minor) comments:

- Sec 3.1. 1) I share the concern from authors that "object detector" should be not confused with "object recognition" (or commonly used "image classifier"). Hopefully vision community could use more consistent terms across literatures.

- Sec 3.1 1) While detection dataset would surely have more scale variation (and truncation / occlusion due to many objects are not in the center), it is not entirely clear that object in detection datasets "vary more in parameters such as lighting, ..., and blur".

- Sec 3.1 1) It would be great to see more analysis on detection datasets (authors mentioned they will discuss in section 4, but only with very little analysis).

- Sec 3.1, 2) While ImageNet and ObjectNet have distinct characteristics, having some stats on object size / spatial placement might better illustrate the gaps between these datasets.

- Sec 3.1 3) Agree top-5 might make classifiers' life easy, but it is more of an eval metric rather than training loss (the model still need to predict top-1 class correct during training). Meanwhile, it is not very clear why multi-label annotation would bias against model that "is spatially more precise but misses some objects". That model should be evaluated against detection benchmarks, rather than object recognition (image classification) datasets.

- Sec 3.1 "bounding box annotation". For multiple objects nearby with the same category, the annotation would include all of them in one bounding box. This might leads to bad aspect ratio? (in general, bounding box would also vary more in aspect ratio than images, and feeding bounding box into a square CNN seems to be less ideal)

- Sec 3.1 "object recognition results". "AlexNet, VGG-19, and GoogLeNet have also been used", GoogLeNet should be "ResNet-152"? ObjectNet uses inception (GoogLeNet) v4 while authors use v3.

- Sec 3.2.1 "The higher the prediction accuracy, the higher the robustness against natural distortions". This is not necessarily true. Looking at Figure 3, it seems all models and both image / bounding box schemes would have decayed performance w.r.t distortion severity, and their slopes seem similar.

- Sec 3.3 "Despite a large body of literature on whether and how much visual context benefits CNNs, the majority being focused on model accuracy, the matter has not been settled yet". "the matter" means context for robustness? please clarify.

- Sec 3.3 " Around 35.5% and 12.6% of the image pixels fall inside the object bounding box and the foreground object, respectively. Around 58.5% of the bounding box pixels fall inside the object mask.". here foreground object and object mask should be the same thing? if so, the numbers seem not matching (would expect box-to-image ratio * mask-to-box-ratio = mask-to-image ratio, or these numbers are normalized differently)?

- Figure 5: it is clear that seg-mask actually isn't robust to adversarial attacks (accuracy dropped significantly), which contradicts with the claim from authors.

- Sec 3.3 "An attempt was made to tune the parameters to attain the best test accuracy in each case", could authors elaborate?

- Sec 3.3.1. here the results seem to indicate segmask is more robust (less variations). Would the segmentation mask itself be indicative for object categories? It might be interesting to predict object classes directly from those masks as a baseline.


Final review:

The authors updated the manuscript and removed tuning experiment on ObjectNet. I am still a bit concerned about the definition of "robustness", but the paper overall does look good for ICLR publication.

---

> ### Author Response · Authors · 2020-11-12
> **Regarding dataset license and answers to your comments**
>
> Regarding dataset license: We double checked (and also communicated with Barbu et al) and confirmed that ObjectNet license indeed does not allow training any model on even a small fraction of it for any task! To address this issue,  we are going to repeat the experiments on a subset of the COCO dataset. Results will be included in the paper before the rebuttal period (for you to see). Meanwhile, our answers to your other comments are given below.
>
>
> Terminology: Computer vision community carefully and correctly uses these terms. ObjectNet paper used the wrong terminology.
> We commented on it in the paper.
>
> Your comment on "vary more in parameters such as lighting, ..., and blur" is valid and well-taken. We will remove it.
>
> Applying recognition models to detection datasets: We mentioned that we will discuss (not promising more analysis). We proposed a new task which is recognizing isolated objects in detection datasets which is different than object recognition and detection tasks in computer vision. Unfortunately, we could not test this on ObjectNet dataset due to its unusual license! Training and testing models on current object recognition datasets including isolated objects does not allow studying context surrounding an object. Including yet another dataset in the paper would make it cluttered and may dilute our message.
>
>
> Sec 3.1, 2) While ImageNet and ObjectNet ...     Response: We will provide a spatial map and distribution of object size, width and hight of ObjectNet objects (notice not all objects in an image are labeled) in the supplement and will put it in perspective with ImageNet dataset.
>
> Meanwhile, it is not very clear why multi-label annotation ....    Response:  Which one is better, a model that achieves x% multi-label accuracy and is spatially accurate or a model with x% accuracy which is less spatially accurate? Multi-label accuracy does not address this. We are just criticizing this measure (it is proposed in Shankar et al., ICML 2020).
>
>
> Sec 3.1 "bounding box annotation". For multiple objects nearby ...  Response: This just happens for few classes (e.g., chair and banana) and few number of images. The images in the ImageNet dataset (over which models are trained and are used here against ObjectNet) also includes images with multiple objects from these classes. That is why we also do the same when annotating objects. In large scale datasets you find all sort of images with multiple or single objects and they are all resized when feeding to a CNN. Despite this, they still generalize well when aspect ratio is distorted at test time (mainly because models rely more on texture).
>
>
> Sec 3.1 "object recognition results". "AlexNet, VGG-19, and GoogLeNet have also been used" ...     Response: Thanks for the correction (will be fixed). We will also mention the difference in inception version (we use pytorch which offers v3). Using v4 will likely make our results even better!
>
>
> Sec 3.2.1 "The higher the prediction accuracy, the higher the robustness against natural distortions". This is not necessarily true ...   Response: This is a very difficult question to answer. The community is still trying to figure/define accuracy vs. robustness. A recent work (Taori et al, 2020) has tried to address this by proposing relative robustness but ideas like these are new and not well established. To illustrate the difficulty consider the following question. Which one of these too models is more robust?! a model with 50%, 30% and 10% accuracy at three increasing levels of distortions or a model with 10%, 10% and 10% at those distortions?!  One might argue the second one is more robust. Nevertheless, since our focus was comparing bounding box vs. full image, we only focus on that and will discard this.
>
>
> Sec 3.3 "Despite a large body of literature on whether ... please clarify ...
> Response:   Both (with more emphasis on adversarial robustness). We will rephrase the sentence!
>
>
>
> Sec 3.3 " Around 35.5% and  ....           Response: Thanks for catching this. It is clearly an error which we address on the new dataset when redoing this analysis.
>
> Figure 5: it is clear that seg-mask actually isn't robust  ...      Response: Can you please double check? The plot clearly shows that seg-mask leads to significantly higher accuracy than bounding box and full image. We follow the tradition in adversarial robustness research.
>
> We used early stopping and tried different optimization methods when training the ResNet-18 model (will be mentioned).
>
>
> Sec 3.3.1. here the results seem to indicate segmask ...
> Response: It is not that less variation makes a model more robust (It is about higher accuracy at those distortions). Segmentation mask (binary mask) will likely be predictive of the object class above chance (and better than bounding boxes) but will perform much lower that segmented object since texture is gone! why would this be an interesting baseline!?
>
> Thank you very much for the thorough review.

---

> > ### Author Response · Authors · 2020-11-19
> > **Regarding your main concern and dataset license**
> >
> > We have now redone the experiments 3.3 using 10 object classes from the MS COCO dataset. Results are different but qualitatively behave the same meaning that segmentation mask leads to higher robustness to adversarial examples, noise corruptions as well as geometric transformation. We have also included some stats on bounding box size in the Appendix H. The new manuscript is uploaded. Your other comments are embraced and reflected as well. Thank you.

---

### Official Review · AnonReviewer4 · 2020-10-27
**Borderline - useful analysis of real-world object detection but not particularly novel**

**Rating:** 6
**Confidence:** 4

**Review:**

Real-world applications frequently provide challenges that are not seen in common computer vision datasets like Imagenet, where images are blurry, dark, corrupted, objects are highly occluded, test objects may be out of distribution due to natural distribution shifts, etc.  This phenomenon was investigated in 2018 by Beery et al, who similarly found categorization is easier with localization in challenging real-world scenes (ie classifying cropped boxes). I would recommend taking a look at that paper (citation below) and including it in the related work.

Beery, S., Van Horn, G., & Perona, P. (2018). Recognition in terra incognita. In Proceedings of the European Conference on Computer Vision (ECCV) (pp. 456-473).


Pros:

The authors provide many experiments digging into various aspects of what makes real-world object detection challenging. This is a useful reference point for future work.

Cons:
This paper re-analyzes generalization in the context of an existing dataset. There is not anything particularly novel about the analysis, and similar results have been shown on other real-world datasets.
The authors describe the image distortions they consider to be “natural”, but are applying them synthetically. It is not clear to me that applying a synthetic distortion of a type that can be seen in the real world is necessarily reflective of those realistic distortions in the wild.  It would be better to explicitly collect examples of these types of distortions in real data and compare against that.

---

> ### Author Response · Authors · 2020-11-11
> **Novel finding that using segmentation masks leads to better models in terms of accuracy and robustness and revisit of prior study**
>
> Thanks for your review and also mentioning Beery et al.'s work with is very related to our work. We will surely discuss it in the paper. They propose a large dataset of animals imaged in the wild in different locations and also, in accordance to our results, show that using boxes leads to lower classification error than the full image. Here, we go one step further and show that using the segmented object is even better which is non-trivial.
>
> Regarding considering  image distortions as "natural". Are you referring to the title in subsection 3.2.1? We agree with you and change the title to "common image corruptions". Well, we were interested in testing whether bounding box is better than full image when dealing with these image corruptions. Testing models against the natural versions of these corruptions in real world (e.g., natural blur or fog) goes beyond our work and we are not aware of such datasets in large scale.
>
> Thanks,

---

### Official Review · AnonReviewer3 · 2020-10-28
**Interesting idea and analysis but not ready yet**

**Rating:** 5
**Confidence:** 4

**Review:**

OVERVIEW:
The authors present a follow-up to the prior work of Barbu et al on the task of Object Recognition* (name confusion addressed in cons below). Barbu et al demonstrated that on a more realistic dataset like ObjectNet, models trained on a clean dataset like ImageNet suffer significant degradation. This work reduces the performance gap by cropping out the object using bounding box or mask information and running the recognition model on top of it. They do this for a variety of models (AlexNet, VGG-19, ResNet-152, Inception-v4, NASNet-A, PNASNet-5L) and transformations (image distortions, adversarial perturbations, context, geometric transformations).

PROS:
- The paper is well-written and tackles an important topic of object recognition* in the wild. It tries to move away from the ImageNet driven approach that is currently present in the community to a more realistic scenario.
- They build on the prior work of Barbu et al and are able to reduce performance gap demonstrated by Barbu et al by using bounding box or mask cropped images of the object of interest.
- They present a lot of experimental evaluation using a variety of models and transformations and demonstrate that their findings hold across all settings.

CONS:
- I agree with the authors that ImageNet with a single (or few) object present in the center of the image with clear foreground-background separation is unrealistic. ObjectNet is a better snapshot of the real world and models trained on ImageNet suffer in ObjectNet. However, the proposed approach to crop out the object using bounding box information or mask information is moving the data distribution from the real world setting of ObjectNet closer to the ideal setting of ImageNet. This then leads to an expected improvement in performance. I appreciate the thoroughness of the results and evaluation presented but it does not feel like a novel contribution in my opinion.
- The authors present 4 future research directions in Section 4. I would be more willing to accept the paper if one of these research directions is incorporated as a contribution. For example, the last research direction of applying an object detection model trained with MS COCO on ObjectNet images instead of an image classification model trained with ImageNet is something that is doable. I would encourage the authors to even consider an object detection model trained on LVIS which has a larger number of object categories. This moves further away from object recognition* to object detection + classification but the latter is what we would typically encounter in a real-world scenario. Even here, I would need some contribution or novel analysis besides re-running current experiments with detection models.
- I strongly recommend a change of name to "Contemplating Real-World Object Classification" (no caps and classification instead of recognition). In my understanding, object recognition is a super-set of classification, detection, segmentation, etc. ImageNet leads to Image Classification models even if they are technically object classification. But sticking with the terminology of Object Recognition because it was used by Barbu et al is misleading. I would prefer Object Classification be used because that is the task of interest in this work.

REASON FOR RATING:
I think there is an interesting problem of real-world object classification that is of significant importance and this work moves a little closer to analyzing possible ways to reduce the performance gap from ImageNet to ObjectNet. However, their key contribution is somewhat expected (not novel) and needs some more work before being conference-paper ready.

UPDATE:
I have read the author feedback and the other reviews/discussions. I keep my original rating of 5. I think multiple authors raised the question of novelty relative to Barbu et al and the authors argue that they demonstrate the importance of context (whole image vs bounding box vs instance mask) for object recognition. Section 3.3 and Figure 5 is helpful in demonstrating it. However, the experimental setup is very limited (700 train + 300 test). COCO has 110K train and 5K val images and many more objects. If you argue that only 10 categories are common between COCO and ObjectNet, how many are common between LVIS and ObjectNet? I would strongly encourage the authors to leverage these pre-trained models and sharpen their message & contributions. I think they provide empirical justifications (important to the community) for expected results in moving from image to bounding box (same comment from multiple reviewers) but they need to de-emphasize that aspect and emphasize their results on context and robustness. A revision and resubmission to a different conference is encouraged.

---

> ### Author Response · Authors · 2020-11-11
> **Comprehensive analysis and clear findings**
>
> Regarding contribution: First, the original ObjectNet paper attracted a lot of attention as it showed a huge performance drop of models. Regardless of improvement using bounding boxes (instead of the full image) being expected or not, it is crucial to fix a flaw which went unnoticed during NIPS review of that paper. Second, as discussed in our paper, whether and how context surrounding an objects impacts accuracy and robustness in object recognition (using CNNs) is still a matter of debate in computer vision and also in human vision. Our finding that using segmented objects leads to higher accuracy and robustness across different image transformations and perturbations is non-trivial and has huge implications of how CNNs are used currently (Did you expect that?!).
>
> Your comment on object detection:  There are three reasons why we did not include object detection here a) To keep the paper focused on object recognition/classification. Including detection would have made the paper more cluttered and there was really no space left, and b) We were not 100% clear whether running object detection models on ObjectNet is a good idea. ObjectNet contains mainly indoor objects. If you look at COCO images, some are indoor and some are outdoor. There are a lot of small objects and lots of objects are occluded. So, COCO might actually be a better dataset to test the object detection models (i.e., COCO images are a good representative of the world for object detection). So, we predict detection models, trained on COCO or OpenImages, to perform well on ObjectNet. One problem though is that there are not many categories in common between ObjectNet and COCO (around 20). These are why we proposed this approach for future.
>
> Regarding changing the title: The term "object recognition" has been heavily used in the community over many years not only in computer vision but also in cognitive vision. It refers to recognizing isolated objects and the term "scene understanding" is usually used to refer to a set of vision problems (recognition, detection, segmentation, etc). Nonetheless, we agree with you that "classification" is a better term here and we will change the title according to you suggestion.
>
>
> Insights: Our work offers important insights regarding object recognition in real world and moves away from intense focus on ImageNet. Instead of keeping the tasks, classification, detection, segmentation, context, etc, separate we attempt to see them as part of a unified solution. These are currently addressed separately in computer vision but are part of a complex unified system in human vision.

---

### Official Review · AnonReviewer1 · 2020-10-30
**Paper addresses Robustness of object recognition pipelines to distribution shift do to natural and synthetic variations.  Interesting paper, but writing quality could be improved.**

**Rating:** 6
**Confidence:** 4

**Review:**

I like the main ideas articulated in the paper, but find the writing lacks some clarity:

Summary of paper: The paper  takes as a starting point the study from Barbu et al where the robustness of object recognition pipelines to be able to handle distribution shifts are studied by testing ImageNet trained architectures against ObjectNet.  The main point in the current paper is that the performance degradation seen in Barbu et al is due to the fact that the CNNs were processing the image with entire image as context  and when one only provides a sub-window around the objects of interest the resulting performance improves significantly.  The paper also describes experiments with various synthetic distorted data and finally examines details of ObjectNet dataset to illustrate that there are images that are hard to categorize even for humans. Thus, the paper concludes that object recognition on ObjectNet is still hard to solve.

It is clear that by using bounding boxes or even removing background from those bounding boxes,  the performance will be better (since the training was on ImageNet with single objects). So, in a way, they are kind-of recreating the training distribution in order to improve the performance.

Main significance of the paper (Pros): Detailed study of performance of object recognition and the empirical finding that figure-ground segmentation may improve recognition.  Analysis of properties of ObjectNet and its challenges.

Originality/Novelty:  The paper is largely empirical and has a good discussion of the relevant background literature analyzing object recognition systems.

Cons:  It has incremental insights.

Clarity of paper: The description of the experiments are at times unclear.    The structure of the paper could be simplified with a table or diagram that illustrates the logic behind the experimentation and conclusion.  There are multiple datasets used, the training scheme is sometimes on ImageNet and tested on ObjectNet and sometimes on selected categories of ObjectNet and test on the rest of ObjectNet.

---

> ### Author Response · Authors · 2020-11-11
> **Important insights**
>
> Thanks for your interest in our work and your comments.
>
> Regarding novelty: We have addressed a very fundamental yet unanswered question: Whether and how much context surrounding an objects impacts accuracy and robustness in object recognition (using CNNs). We also extensively discuss and relate object recognition, detection, and segmentation (in particular model evaluation and scores). Our finding that it is better to train models on segmented masks rather than bounding boxes is non-trivial and very important moving forward in object recognition, akin to how humans process scenes and recognize objects. We hope our results will encourage researchers to explore these ideas further.
>
>
> The reason to use multiple datasets is to show that proposed solutions generalize across datasets and are not limited to only one dataset.  Further, the choice of data in different sections is partly due to restrictions imposed by the ObjectNet license. In fact, we are going to replace the analysis in section 3.3 using the COCO dataset since ObjectNet license prohibits training ANY model on even fractions of it!
>
> Also, notice that we have annotated the bounding boxes of around 300 classes from the ObjectNet dataset and plan to extend it to all 500 classes. We will be sharing the annotations with the public.
>
> We will improve the writing in the final version of the paper according to your suggestions.
>
> Thank you.

---

### Author Response · Authors · 2020-11-19
**An issue regarding dataset license brought up by AnonReviewer2**

AnonReviewer2 raised a concern regarding dataset license. Dataset license prohibits any use of the dataset for training models in any way, shape, or form! We had a small experiment that violated that. In corresponding with dataset creators, they confirmed the restriction. Therefore, we repeated that experiment using another dataset. Our conclusions and observations remain the same.

We have also taken into account all 4 reviewers' comments and have updated the manuscript correspondingly.

Thanks for your comments.

---

### Decision · Program_Chairs · 2021-01-07
**Final Decision**

**Decision:**

Accept (Poster)

**Comment:**

Reviewers agreed that overall the two-pronged message of the submission has utility.

1. That ObjectNet is continues to be difficult for models to understand and is a challenging test platform even when objects are isolated from their backgrounds. This is significant and not obvious. Cropping objects makes the distribution shift between ObjectNet and ImageNet far smaller, but the large remaining performance gap points to the fact that detectors are limited by their ability to recognize the foregrounds of objects not by their ability to isolate objects from their backgrounds.

2. That segmentation could be a promising direction for robustness to adversarial perturbations which has so far been overlooked.